# Annotating pathogenic non-coding variants in genic regions

Sahar Gelfman [1,2], Quanli Wang[1,2], K. Melodi McSweeney[1,2], Zhong Ren[1,2], Francesca La Carpia[3], Matt Halvorsen[1,2], Kelly Schoch[4], Fanni Ratzon[5], Erin L. Heinzen[1,3], Michael J. Boland[1,6], Slavé Petrovski [1,7] & David B. Goldstein[1,2]

Identifying the underlying causes of disease requires accurate interpretation of genetic variants. Current methods ineffectively capture pathogenic non-coding variants in genic regions, resulting in overlooking synonymous and intronic variants when searching for disease risk. Here we present the Transcript-inferred Pathogenicity (TraP) score, which uses sequence context alterations to reliably identify non-coding variation that causes disease. High TraP scores single out extremely rare variants with lower minor allele frequencies than missense variants. TraP accurately distinguishes known pathogenic and benign variants in synonymous (AUC = 0.88) and intronic (AUC = 0.83) public datasets, dismissing benign variants with exceptionally high specificity. TraP analysis of 843 exomes from epilepsy family trios identifies synonymous variants in known epilepsy genes, thus pinpointing risk factors of disease from non-coding sequence data. TraP outperforms leading methods in identifying non-coding variants that are pathogenic and is therefore a valuable tool for use in gene discovery and the interpretation of personal genomes.

[1] Institute for Genomic Medicine, Columbia University Medical Center, New York, New York 10032, USA. [2] Department of Genetics and Development, Columbia University Medical Center, New York, New York 10032, USA. [3] Department of Pathology and Cell Biology, Columbia University Medical Center, New York, New York 10032, USA. [4] Department of Pediatrics, Duke University Health System, Durham, North Carolina 27705, USA. [5] Department of Pathology, Lenox Hill Hospital, New York, New York 10075, USA. [6] Department of Neurology, Columbia University, New York, New York 10032, USA. [7] Department of Medicine, Austin Health and Royal Melbourne Hospital, University of Melbourne, Melbourne, Victoria 3050, Australia. Correspondence and requests for materials should be addressed to S.G. (email: sahar.gelfman@columbia.edu)

Major advances in sequencing technologies in recent years have made next-generation sequencing the primary tool for identifying causal variants in rare diseases. Recent studies have reported causal variants in multiple diseases including epilepsy[1], Alzheimer's disease[2], congenital heart disease[3] and ALS[4, 5].

With the accessibility of sequence data, the focus now shifts to accurate data interpretation. Several approaches have been proposed to identify causative variants from sequence data[6–8]. Pinpointing the causative variant requires filtering and prioritizing gene variants. Sequencing of large populations allows filtering out common alleles that are much less likely to cause rare diseases. Large population data sets such as: ExAC[9] (60,700 exomes), EVS[10] (6,500 exomes) and Kaviar[11] (13,200 whole genomes and 64,000 exomes) allow filtering out the common alleles, while retaining rare alternative alleles. Rare allelic variants are then prioritized based on their predicted ability to result in protein damage. Numerous tools and scores are available for this purpose, predicting effects based on amino-acid substitutions (PolyPhen-2[12], SIFT[13], FATHMM[14]), conservation (GERP++[15]) or an ensemble of annotations and scores (CADD[16], MutationTaster[17], GWAVA[18]). These approaches predict functional coding variants with very high accuracy, in some measure due to the high conservation of protein sequences. However, variants that do not change the amino-acid sequence, such as intronic and synonymous variants, are under lower evolutionary constraints[19, 20], making them much harder to prioritize using these tools and resulting in these variants being mostly discarded in genome interpretation analyses. Yet non-coding variants, while not necessarily under as strong evolutionary constraint, can potentially have deleterious effects on a transcript through the regulation of splicing or transcription in a species-specific manner[21]. Therefore, successfully capturing these indirect effects will enable prioritizing the magnitude of damage a synonymous or intronic variant will cause.

The Transcript-inferred Pathogenicity (TraP) score, presented here, is constructed to evaluate a single nucleotide variant's ability to cause disease by damaging a gene's transcripts and subsequently also its protein products. To ensure that TraP captures signals unrelated to amino-acid substitutions, the model was trained only on synonymous variants. TraP was further evaluated using only variant data sets of either intronic or synonymous annotations. Through a comprehensive evaluating scheme, we demonstrate TraP's negative correlation with allele frequency using population sequence data, and present TraP's extreme specificity when distinguishing known pathogenic and benign, synonymous and intronic variants. TraP's application to sequence data proves successful in identifying known risk factors of epilepsy from patients' de novo mutations, as well as pinpointing a specific disease gene in a family-trio exome-sequencing study. Considered together, this evaluation approach exhibits TraP's ability to correctly prioritize pathogenicity of non-coding variants when interpreting human genomes.

## Results

**Construction of the TraP score and model evaluation.** The TraP score was constructed using three main components: (1) Information acquisition—details of the harboring gene and its transcripts are gathered for each variant (Fig.1a–1). The GERP++ Rejected Substitutions score[15] (GERP++ score) is also obtained for measuring evolutionary constraints acting on a specific coordinate. (2) Feature calculation—possible changes to sequence motifs are evaluated, including changes to exon–intron boundaries (Fig.1a–2), creation of cryptic splice sites (Fig.1a–3), creations and disruptions of *cis*-acting binding sites for splicing

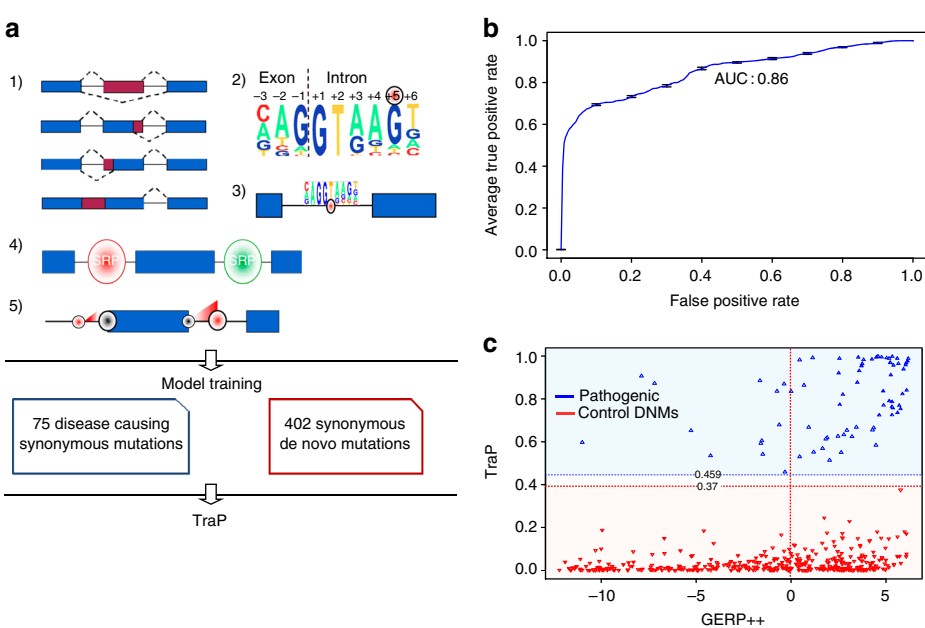

**Fig. 1** TraP model construction and evaluation. **a** TraP construction workflow and main features calculated for TraP: (1) Information acquisition from all genes and transcripts that harbor by the variant, (2) changes to splice site motif that affect it's binding affinity to the splicing machinery, (3) creations of new splice junctions that might interact with the splicing machinery, (4) creations and disruptions of *cis*-acting binding sites to splicing regulatory proteins (SRP), (5) interactions between features, such as a stronger effect of a new splice site on an exon with a weak original splice site (*red representing a new splice site*). Model is trained using synonymous variants that are either known pathogenic variants (*blue box, left*) or DNMs from healthy individuals (*red box, right*). **b** A receiver-operating characteristic curve showing the results of 10 rounds of 10-fold cross-validations with an average AUC of 0.86. **c** Model predictions of the training-set show a clear separation of pathogenic variants (*blue*) versus control DNMs (*red*). TraP (*y-axis*) exhibits a minimum threshold for pathogenic variants of 0.459, below, which reside all control DNMs. GERP++ score (*x-axis*) considers 49.5% of benign variants as conserved

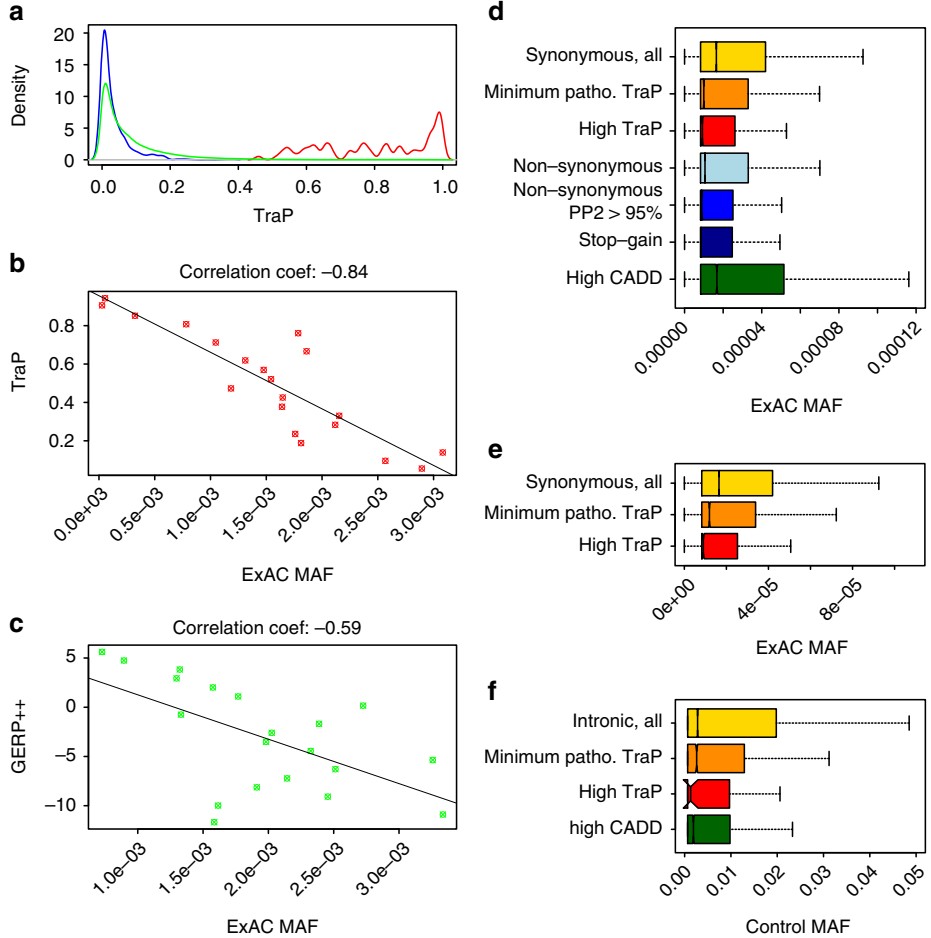

**Fig. 2** TraP and allele frequency of synonymous and intronic variants. **a** TraP density plots for training-set pathogenic variants (*red*), control DNMs (*blue*) and 1.46 M ExAC synonymous variants (*green*). **b** Correlation between TraP and MAF for 29,985 synonymous variants that create strong cryptic splice sites. The data set was binned into 20 groups by taking 5% score intervals and examining the correlation of the 20 points with the average MAF for each group. **c** Correlation between GERP++ score and MAF for 29,985 synonymous variants that create strong cryptic splice sites. The data set was binned 20 groups as in (**b**). **d** MAF distributions for different types of variants. MAF distribution for synonymous variants is presented with no Trap threshold (*yellow*), minimum pathogenic TraP (≥ 0.459, *orange*) and high TraP (≥ 0.93, *red*). Synonymous variants with high TraP (*red*), have significantly lower average MAF than NS variants (*bright blue*). MAF distribution of CADD top scoring synonymous variants (97.84th percentile) is also presented (*green*). **e** MAF distributions based on a non-GERP++TraP model for 1.46 M ExAC synonymous variants. Thresholds used differ from the final TraP model: minimum pathogenic TraP threshold used is the 25th percentile score (≥ 0.66, *orange*) and high TraP threshold is the 75th percentile score (≥ 0.955, *red*). **f** MAF distributions for 1.5 M intronic variants from 776 sequenced whole genomes. MAF distribution is presented for variants with no Trap threshold (*yellow*), minimum pathogenic TraP (≥ 0.459, *orange*) and high TraP (≥ 0.93, *red*). The *whiskers* of the *boxplots* extend to the most extreme data point, which is no more than 1.5 times the interquartile range away from the box

regulatory proteins (Fig. 1a–4), interactions between selected features such as original and new splice site scores (Fig.1a–5) and others (Supplementary Data 1). Overall, 42 features and 14 general properties (chromosome, strand, coordinate, and so on) are collected for each variant. (3) Modeling—the last component of the score construction is the incorporation of selected features into a random forest model. The model is then trained on a set of pathogenic and benign variants and its performance is evaluated.

To train the TraP model we constructed a data set of 75 pathogenic synonymous variants and 402 benign variants (the 'training-set', Fig. 1a). Pathogenic variants were carefully curated from published studies where each variant is strongly associated with rare disease (Supplementary Data 2). We used 402 de novo mutations identified among healthy individuals (control DNMs) as benign variants[22]. Healthy DNMs were used as negative controls as a preventative measure to assure the model did not train to select features that are specific only to rare variants. Using

a set of benign population variants as controls might have trained TraP against more common variants.

The final TraP score produced by the model is in a range between zero and one (0–1) and represents the fraction of decision trees that classify a variant as pathogenic.

The model was tested with different sets of features and re-evaluated for the ability to distinguish training-set pathogenic and benign variants. The final model that has the best performance is using 20 selected features (Supplementary Data 1) and the importance measurements of each feature's contribution were measured (Supplementary Data 3 and Supplementary Information). This model has an accuracy of 91.82% (8.18% out-of-bag error rate). To further test the robustness of the prediction model, we also followed with an elaborate scheme of ten 10% cross-validations by which an average Area Under the Curve (AUC) of 0.86 was achieved, supporting the model accuracy measurement (Fig. 1b).

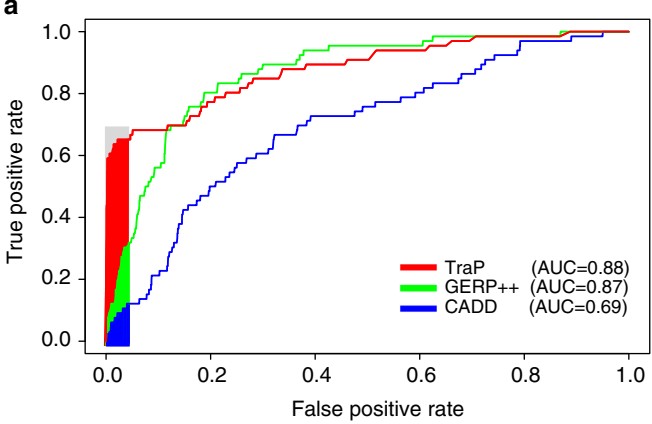

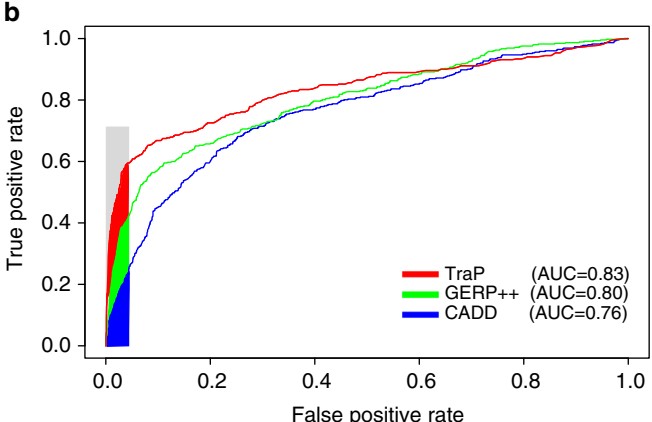

**Fig. 3** ROC curves of ClinVar pathogenic and benign variants. **a** A ROC curve of ClinVar pathogenic and benign synonymous variants, calculated for TraP (*red*), GERP++ (*green*) and CADD (*blue*). **b** Same as **a** but for ClinVar intronic variants. Colored area represents high specificity region

TraP presents a clear-cut discrimination of pathogenic from benign training variants, scoring all pathogenic variants ≥ 0.459 with an average TraP of 0.8. On the other hand, 99% of benign variants are scored below 0.18, and all benign variants fall below 0.37, with an average TraP of 0.034 (Fig. 1c). While TraP shows a clear separation, GERP++ shows ambiguous values: almost half (49.5%) of the control DNMs are considered conserved and almost 20% of the pathogenic variants are located in non-conserved positions (i.e., GERP++ score lower than zero). Thus, while conservation helps TraP identify some pathogenic variants, TraP also identifies the transcript-damaging potential of nucleotide substitutions where the position itself is not under evolutionary constraint.

**TraP properties of variants in the general population**. To assess the general properties of TraP prediction, it was tested using large datasets of synonymous and intronic variants. As a synonymous dataset we used the information from 60,706 exomes (ExAC database)[9]. The ExAC database represents a human population reference cohort that also includes samples from various common disorder studies such as type-2 diabetes, schizophrenia and myocardial infarction. However, individuals affected by severe pediatric disease were removed from the dataset so that it should serve as a useful control set of allele frequencies for severe disease studies. We retrieved all the synonymous variants from ExAC, identifying 1.46 million variants. The average TraP for ExAC synonymous variants is 0.087, which is dramatically lower than the 0.8 average score for the training-set pathogenic variants

(MW-test, *P*-value = $6.4 \times 10^{-50}$), and slightly higher than the 0.034 average TraP for the training-set control DNMs (MW-test, *P*-value = $9.4 \times 10^{-35}$, Fig. 2a). Only 2.16% of ExAC variants (31,520 variants) rank equal to or higher than the minimum 0.459 score threshold for pathogenic variants, suggesting 97.84% are benign. We can expect that this small subset of 2.16% ExAC pathogenic classified variants will be a mixture of clinically relevant variants that reside among ExAC population and also false positives arising from some background noise that can occur due to biological/technical reasons within the ExAC sample. Thus, the ExAC reference cohort sample adopted here informs us that in a large human population sample the expected rate of TraP-based pathogenic-classified synonymous variants is ∼ 2%.

We next evaluated TraP for variants that reside only within introns. For this purpose, we analyzed whole genome sequencing data from 776 genomes that are available for control use. These control genomes produced 18,377,624 intronic variants with high mapping quality. We randomly selected 1.5 M variants for further analysis. The average TraP for intronic variants is 0.069, significantly lower than the score for 1.46 M synonymous variants (MW-test, *P*-value < $1 \times 10^{-100}$). Furthermore, only 0.6% (8,644 out of 1.5 M) intronic variants pass the minimum 0.459 pathogenic TraP (data not shown), highlighting that the expected rate from large human population samples of TraP-based pathogenic-classified intronic variants is ∼ 0.6%. Importantly, the median GERP++ score for Trap-considered pathogenic intronic variants is 0.12, suggesting that half of them are not under evolutionary constraints (Supplementary Fig. 1).

These results suggest that TraP is very selective in introns and further supports that variants considered as highly deleterious by TraP might not be under strong evolutionary constraints.

**High TraP scores correlate with lower allele frequencies**. Among the most sensitive tests of whether a class of variants is under negative selection is the comparison of allele frequencies to variants that are presumed neutral or under less selection. To evaluate the relationship between TraP and minor allele frequency (MAF), we first tested the full set of 1.46 M ExAC synonymous variants. Since both TraP and MAF distributions are highly positively skewed with most values clustered around zero (Supplementary Fig. 2), we binned the dataset into 20 groups by taking 5% score intervals and examining the correlation of the 20 points with the average MAF for each group. We repeated this analysis also for the GERP++ score. We find that GERP++ correlation with MAF is − 0.82 (*P*-value = $4.7 \times 10^{-06}$) and TraP's correlation with MAF is − 0.52 (*P*-value = 0.021).

Based on the TraP model importance table, a 'Cryptic 5' Splice Site Score' (feature F7) has a high contribution in predicting pathogenicity. This class of variants represents variants expected to be particularly sensitive to TraP but not necessarily to conservation. Focusing on the subset of 29,985 synonymous variants that are classed as creating a new, strong cryptic 5'ss (PSSM score > 84), we find that the TraP correlation with MAF is stronger *r* = − 0.84 (*P*-value < $2.2 \times 10^{-16}$) compared to the correlation between GERP++ and MAF of *r* = − 0.59 (*P*-value = 0.0069, Fig. 2b, c). When Sub-selecting cryptic splice site variants that are predicted as pathogenic (TraP ≥ 0.459, 6,328 variants), TraP correlation is − 0.51 (*P*-value = 0.025) while GERP++correlation is − 0.15 (*P*-value = 0.53, Supplementary Fig. 3). These results exhibit how TraP can help identify potential variants that are deleterious to the transcript (thus, strongly selected against in the human lineage), but do not necessarily have a strong conservation signature.

We next compared allele frequencies between synonymous variants that TraP considers as pathogenic against several types of

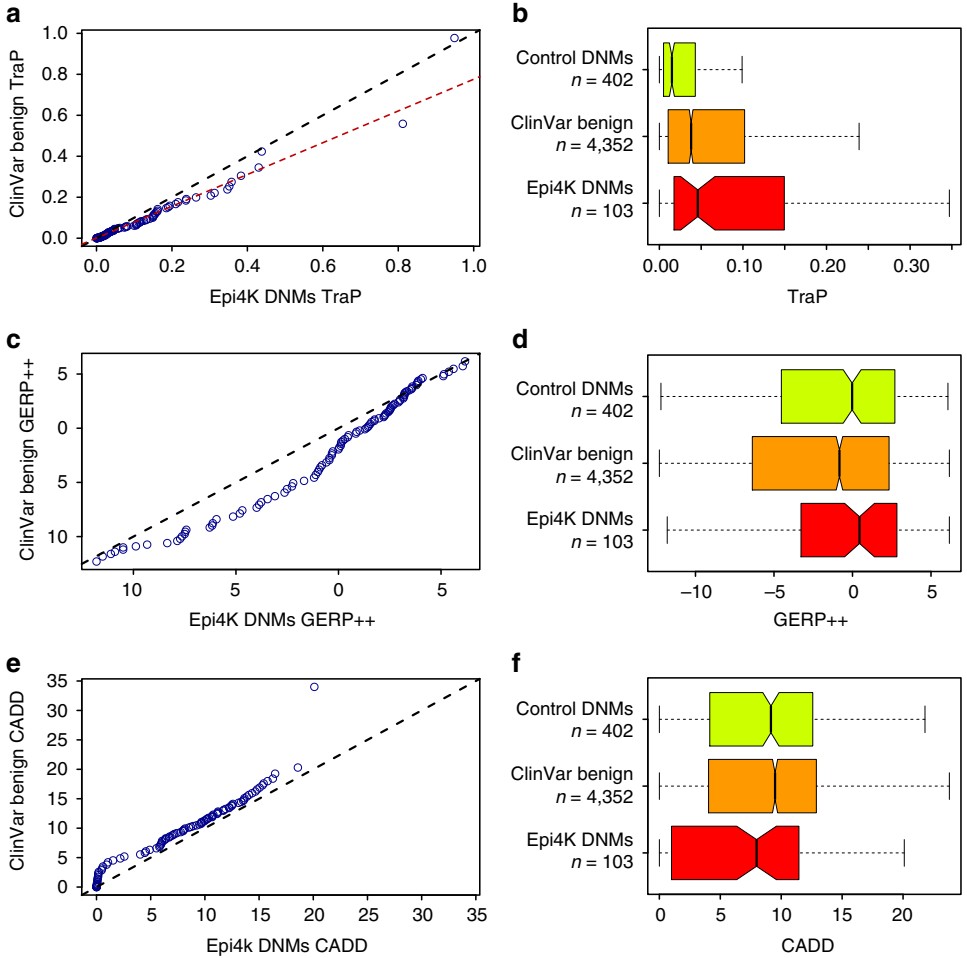

**Fig. 4** Epilepsy synonymous DNMs vs. ClinVar benign controls. A quantile–quantile plot for 103 Epi4K DNMs and 4,352 benign ClinVar synonymous variants is calculated for **a** TraP scores, **c** GERP++ scores and **e** CADD scores. Score distributions for training-set control DNMs, ClinVar benign variants and Epi4K DNMs are scored using **b** TraP, **d** GERP++ and **f** CADD.The *whiskers* of the *boxplots* extend to the most extreme data point, which is no more than 1.5 times the interquartile range away from the box

coding variants retrieved from the ExAC database: all Non-Synonymous (NS) variants, a subset of only Stop-Gain (SG) variants and a subset of missense variants with high (95th percentile) Polyphen-2 scores (sizes of datasets are depicted in Supplementary Data 5). We observe that synonymous variants with higher than minimum pathogenic TraP ($\geq 0.459$, 31,520 variants) and NS variants have a significantly lower MAF than the remaining TraP predicted benign synonymous variants (Fig. 2d, *yellow*, *orange* and *bright blue* columns, multiple MW-tests, *P*-value $< 8.6 \times 10^{-93}$). We also find that a higher TraP threshold results in significantly lower MAF. Strikingly, we find that a TraP threshold of 0.93 and above (5,402 variants) identifies synonymous variants with significantly lower allele frequencies than NS variants (Fig. 2d, *red* and *bright blue* columns, MW-test, *P*-value = 0.046). The lower MAF distributions of high TraP ($\geq 0.93$) synonymous variants than NS variants highlight that this sub-group of synonymous variants appears to be under similar levels of negative selection as the NS class of variants and might be as deleterious to the protein product.

As observed in Fig. 2b, c, TraP adds significant information to conservation for specific classes of variants. We next examined whether TraP can identify rare variants without conservation information. We constructed a TraP model without the GERP++ score as a feature. The model without GERP++ as a feature achieved an accuracy of 90.5% (compared to the original 91.8%

accuracy). We next tested this model using the 1.46 M ExAC synonymous variants. When comparing MAF of high and low TraP variants in the GERP-less model, the results exhibit a similar strong decrease in MAF distribution as seen with the original TraP (Fig. 2e). We find a significant decrease in MAF distribution between all the synonymous variants and higher TraP variants (Fig. 2e, *yellow* and *orange bars*, MW-test, *P*-value $= 4.68 \times 10^{-09}$). In this analysis, a TraP threshold of 0.66 was used to classify minimum pathogenic TraP scores, which corresponds to the 25th percentile of the GERP-less training-set. When further increasing the threshold (TraP $> 0.955$, i.e., the 75th percentile among the GERP-less training-set), the MAF distribution further decreases, and is significantly below the MAF of synonymous variants with intermediate TraP ($\geq 0.66$ and $< 0.955$, *orange* and *red bars*, MW-test, *P*-value $= 2.05 \times 10^{-10}$). These results highlight that the GERP-less TraP captures the relevant signals among possible variants to help identify this class of variants that damage the transcript and are selected against in the human population.

The CADD score, a widely used annotation ensemble score, incorporates many different features such as conservation and regulatory annotations and is used to evaluate coding and non-coding variants[16]. In order to determine how TraP performs in comparison to CADD, we examined CADD scores using the same ExAC synonymous dataset. Specifically, we tested CADD's

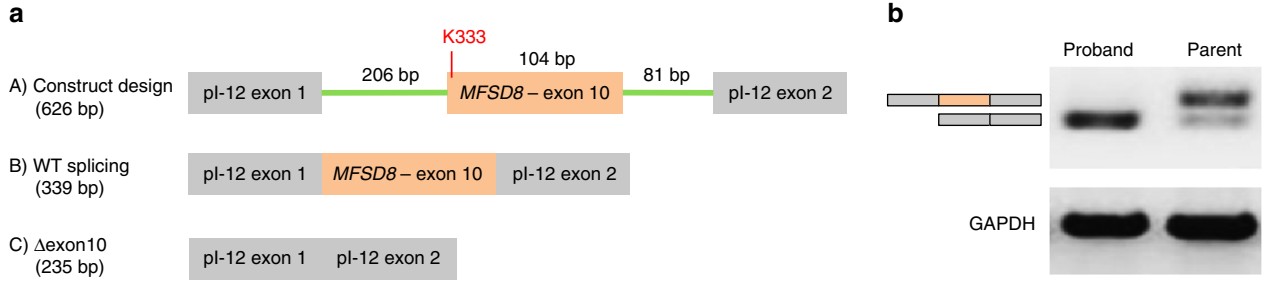

**Fig. 5** Mini-gene design and quantification. **a** Minigene design. (A) Exon 10 and flanking genomic sequence was amplified from patient and parent DNA and cloned into the pI-12 splicing reporter vector. (B) Predicted splicing effect if splice site mutation has no effect on WT splicing. (C) Predicted skipping of exon 10 if splice site is disrupted by K333. **b** Semi-quantitative PCR gel of splicing isoforms of parent harboring the W97C variant and proband harboring both the W97C and K333 variants

relationship with MAF by comparing the MAF of the top 2.16% CADD scored variants (Fig. 2d, *green column*) to the rest (lower 97.84%) of the variants. Surprisingly, the synonymous variants with highest CADD scores have a higher MAF than the remaining ~98% of the variants with lower CADD scores. CADD identifies the more common synonymous variants as more damaging ($P$-value $= 2.28 \times 10^{-112}$). Allele frequencies for top CADD variants are also significantly higher than NS mutations and variants with a TraP score $\geq 0.459$ (multiple MW-tests, $P$-value $< 1.8 \times 10^{-214}$). In addition, CADD considers the known pathogenic variants from the training-set (CADD $12.88 \pm 4.95$) as significantly less damaging than NS variants (CADD $18.67 \pm 9.84$, MW-test, $P$-value $= 9.4 \times 10^{-11}$, Supplementary Fig. 4).

When examining intronic variants' correlation with MAF, the 1.5 M intronic control variants exhibit the same decreasing MAF with increasing TraP scores as observed for synonymous variants (Fig. 2f, *yellow* to *red columns*). Contrary to its results with synonymous variants, high CADD variants exhibit the expected lower median allele frequency in intronic regions (Fig. 2f, *green column*). Taken together, these results show that TraP selects both intronic and synonymous variants that have lower MAF and are under stronger negative selection.

**TraP identification of pathogenic and benign variants**. The relationship between TraP and MAF confirms that TraP reliably identifies variants that are selected against in the population. We next tested how TraP performs in determining variant pathogenicity. For this purpose, we used curated benign and pathogenic datasets obtained from the ClinVar database[23], which is a public archive of the relationships between gene variants and phenotypes. ClinVar sets a clinical significance value to each variant as recommended by the American College of Medical Genetics and Genomics (ACMG)[24]. We obtained two ClinVar datasets for intronic and synonymous variants. For each dataset, we selected only variants that are considered as either 'benign' or 'pathogenic' by ClinVar and scored all variants using TraP.

The dataset of synonymous variants from ClinVar consists of 4,418 variants, 66 are 'pathogenic' that do not overlap with the 75-pathogenic training-set variants; the rest (4,352) are classified as 'benign'. The average difference in TraP scores between the groups is highly significant (0.62 vs. 0.078, pathogenic versus benign variants, respectively, MW-test, $P$-value $= 7.1 \times 10^{-27}$). We next tested the specificity and sensitivity of different TraP score thresholds. When plotting a Receiver Operating Characteristic (ROC) curve using TraP predictions for pathogenic and benign variants, the AUC is 0.88. Using the minimum TraP for pathogenic training variants ($\geq 0.459$) as a cutoff, specificity is a very high 98.4% and sensitivity reaches 63.6% (filled area in Fig. 3a). With TraP above 0.75, specificity reaches 99.7% while achieving sensitivity of 59%. These results exhibit a strong

capability for TraP to identify true negatives, while still pinpointing a large proportion of the pathogenic variants (Fig. 3a, *red line*).

In comparison with TraP, the GERP++ score alone also separates both groups with a significant difference (MW-test, $P$-value $< 6.85 \times 10^{-24}$), and with a similar AUC of 0.866 (Fig. 3a, *green line*). However, to achieve a specificity of 98.4% (as achieved by a minimum pathogenic TraP) a very high GERP++ score is required ($>5.6$) that misses most pathogenic variants with a very low sensitivity of 12%. These results demonstrate that GERP++ alone cannot be used to identify the pathogenicity of synonymous variants.

CADD scores for pathogenic variants are also significantly higher than benign variants (Fig. 3a, *blue line*, MW-test $P$-value $= 2.8 \times 10^{-08}$, AUC is 0.69). To reach the minimum pathogenic TraP specificity of 98.4%, CADD also misses most pathogenic variants with a true positive rate of 6% above a 19.6 CADD threshold.

The ClinVar data set of intronic variants consists of 3,266 variants, 452 are considered 'pathogenic' and 2,814 are 'benign'. Average TraP scores are highly significant between the groups: 0.52 vs. 0.11, pathogenic versus benign, respectively (MW-test, $P$-value $< 8.32 \times 10^{-114}$), and the AUC is 0.83, suggesting a successful identification of pathogenicity (Fig. 3b, *red line*). The minimum pathogenic TraP score specificity is 97.1%, almost as high as synonymous classification, while sensitivity reaches a comparable 56.6% (filled area in Fig. 3b).

When interpreting a patient's genome in search for pathogenic variants, especially within intronic regions, ruling-out false positives is of utmost priority. We therefore tested specificity using a higher TraP threshold. We find that a TraP score above 0.75 provides very high specificity (99.4%) while managing a sensitivity of 35.8%.

Testing the GERP++ score of intronic variants, we find that pathogenic and benign variants are significantly different (MW-test, $P$-value $< 3.3 \times 10^{-94}$) with an AUC of 0.8 (Fig. 3b, *green line*). To achieve a specificity of 97.1%, as achieved by a minimum pathogenic TraP, a very high intronic GERP++ score is required ($>4.65$) with a sensitivity of 38% (compared TraP's 56.6%). Reaching 99.4% specificity, as gained by 0.75 TraP thresholds, drops the GERP++ sensitivity to 11.7%.

When comparing CADD's ability to differentiate pathogenic and benign intronic variants, AUC is 0.76 and groups are significantly different (Fig. 3b, *blue line*, MW-test, $P$-value $< 5.8 \times 10^{-71}$). However, to reach a high specificity of 97.1% as with minimum pathogenic TraP, the CADD score has to be higher than 16.85. This score has a sensitivity of 18.8%, missing two thirds of the variants caught using TraP and half the variants caught using GERP++. A 99.4% specificity, as achieved by a 0.75 TraP, drops CADD sensitivity to 8.2%.

**Table 1 Epi4K DNMs with high TraP scores**

| Variant ID | TraP | Gene ID | Phenotype associated with Gene | Ref |
|---|---|---|---|---|
| **15-23062273-T-G** | **0.949** | ***NIPA1*** | **Hereditary spastic paraplegia and association to Epilepsy** | 29 |
| 18-40695437-G-A | 0.812 | RIT2 | Schizophrenia, synaptic signaling; weak PD | 31, 32 |
| 2-222428815-G-A | 0.438 | EPHA4 | *ALS, neurogenesis; strong interaction with* ARHGEF15 (*a known epilepsy gene*) | 33 |
| 14-23858626-T-C | 0.430 | MYH6 | Cardiomyopathy | |
| 3-42700087-G-A | 0.383 | ZBTB47 | Unknown | |
| **15-93472268-C-T** | **0.358** | ***CHD2*** | **Epileptic encephalopathy** | 27 |
| 13-52249316-A-G | 0.352 | WDFY2 | Unknown | |

Genes associated with epilepsy are in bold and genes associated with other neurological disorders are underlined.

Taken together, these results demonstrate the strength of TraP in identifying both intronic and synonymous pathogenic variants. Indeed, when requiring high specificity, TraP identifies three to four times as many pathogenic variants as CADD or GERP++. The reason that high conservation misses many of the pathogenic variants might once again lie in variants that reside in un-constrained non-coding positions (see Discussion).

**TraP identifies risk factors of epilepsy**. As demonstrated above, TraP can identify pathogenic synonymous and intronic variants that are mostly overlooked in exome interpretations. Theoretically, the addition of TraP-defined deleterious variants that were previously dismissed in genetic analyses might implicate a gene as a risk factor where it was previously unnoticed. We therefore used TraP to search for risk candidates of synonymous origins. For this purpose, we used de novo variants collected from exome sequence data encompassing 281 epilepsy family trios sequenced by the Epi4K consortium[25, 26]. Overall, 103 synonymous de novo mutations (Epi4K DNMs) were collected using stringent criteria (Supplementary Data 6, see Methods). TraP scores were calculated for the 103 Epi4K DNMs and were compared to the 4,352 synonymous benign variants already obtained from ClinVar (used for AUC calculations in Fig. 3a). We reasoned that it is plausible Epi4K DNMs might not have any damaging effects or epilepsy association, while on the other hand ClinVar benign variants might harbor a small fraction of damaging variants not yet associated with disease. Thus, TraP scores for the two datasets are expected to overlap to an extent. However, a tendency towards high TraP scores in epilepsy cases could point to genes that might be implicated as risk factors in epilepsy.

We compared the two datasets using a non-parametric quantile–quantile plot approach while interpolating the values of the larger data set to fit the size of the smaller one. Comparing TraP of Epi4K DNMs to ClinVar benign variants shows a shift toward higher scores in the epilepsy cohort, specifically for TraP scores that are higher than 0.3 (Fig. 4a, *dashed red line*). Furthermore, TraP scores for Epi4K DNMs (Fig. 4b, *red box*) are significantly higher than ClinVar benign variants (Fig. 4b, *orange and red boxes*, MW-test, P-value = 0.015).

Surprisingly, Epi4K DNMs have higher GERP++ scores mostly in the lower, not conserved percentiles (Fig. 4c), rendering high GERP++ scores uninformative for identifying risk candidates in the Epi4K dataset. Moreover, GERP++ score distribution is similar between Epi4K DNMs and the training-set control DNMs (Fig. 4d, *yellow and red boxes*).

CADD exhibits the opposite trend, providing higher scores for ClinVar benign variants than the Epi4K DNMs dataset (Fig. 4f, MW-test, P-value = 0.0072), suggesting the Epi4K DNMs dataset is less damaging than population benign variants.

The fact that de novo mutations identified in patients with epilepsy have higher TraP scores than those seen in individuals without such a diagnosis suggests that these variants might influence disease risk via effects on the transcript. We therefore examined all the Epi4K DNMs that pass a threshold score of 0.35, which is within the score shift region observed in Fig. 4a, and almost five times higher than a benign average score. There are 7 DNMs that pass the threshold score (Table 1). Two of these variants reside in the *NIPA1* and *CHD2* genes. The latter is a known risk factor for epilepsy[27]. *NIPA1* is not considered a definitive epilepsy gene, yet has accumulating recent evidence for its association with epilepsy[28–30]. This analysis implicates these variants as strong epilepsy candidates. Of the remaining 5 genes with high TraP, two more are associated with other neurological diseases through synaptic signaling and neurodegeneration (*RIT2*[31, 32] and *EPHA4*[33]).

The total number of human genes that are associated with epilepsy differs between reports, ranging from 73 strongly associated[34] to 270 that are implicated to some extent in the literature[35]. When considering even a lenient measure for epilepsy association: 270 epilepsy genes out of 20,686 human protein coding genes (1.3%), we find the gene list in Table 1 (2/7 or 28.6%) to be significantly enriched for epilepsy associated genes (Fisher's exact test, P-value = 0.0035) demonstrating that TraP identified epilepsy risk factors within the Epi4K DNMs. It further suggests that other genes from Table 1 might also harbor risk for epilepsy and should be further examined using burden analyses and tested for existence in other epilepsy cohorts.

**Application of TraP in a single case study**. Here we report a specific case- and unaffected parents-trio analysis that, by application of TraP, allowed the identification of a causal synonymous variant.

The patient reported is a 15-year-old female of Caucasian ancestry. The patient presented with progressive vision loss with decreased color and night vision, and retinal dystrophy at 11 years old. Exome sequencing of proband and parents was done when she was 13 years old and revealed compound heterozygous variants in the Major Facilitator Superfamily Domain Containing 8 (*MFSD8*) gene (OMIM, 611124), which include a novel non-synonymous mutation W97C (NM_152778.2:c.291-G > C) and a novel synonymous mutation K333 (c.999 G > A) at the first nucleotide of exon 10 (position 1 of the 3′ splice site). Mutations in *MFSD8* are associated with Neuronal Ceroid Lipofuscinosis (NCL), a progressive neurodegenerative disorder characterized by the intracellular accumulation of auto-fluorescent lipo-pigment storage material in the brain. Phenotypes include progressive dementia, seizures and progressive visual failure. Because of the patient's overlapping ophthalmologic symptoms, and the autosomal recessive mode of inheritance of NCL, *MFSD8* was suspected to be the causal gene for the patient's symptoms.

The missense variant W97C is putatively damaging to the protein because 1) it is not found in >60 thousand individuals in the ExAC database, and therefore is considered extremely rare, and 2) it is predicted as 'deleterious' and 'possibly damaging' to the protein by both the SIFT and Polyphen-2 scores, respectively.

TraP score of the K333 synonymous variant was calculated as 0.58, well into the range of having a pathogenic effect. We therefore evaluated K333's effect on splicing of the *MFSD8* transcript. Exon 10 of *MFSD8* and flanking intronic sequences were amplified from genomic DNA isolated from the patient and the parent lacking the synonymous mutation. The region was cloned into the pI-12 splicing reporter vector and over-expressed in HEK293T cells (Fig. 5a). Semi-quantitative PCR demonstrates that the K333 mutation results in aberrant splicing of the *MFSD8* transcript—inducing the full exclusion of exon 10 (Fig. 5b). Sanger sequencing confirmed full skipping of exon 10 in all replicates with the K333 mutation. The wild-type construct also exhibits a small amount of the skipping isoform; however, the wild-type isoform predominates (Fig. 5b). These results present strong evidence for K333 resulting in damage to MFSD8. Taken together, both variants (missense and synonymous) have deleterious effects on both copies of *MFSD8*, thus fitting the autosomal recessive mode of inheritance of NCL, which requires two aberrations to both copies of the gene in order for the disease to develop. At age 15 years, after diagnosis using exome sequencing, the patient began having generalized tonic-clonic seizures and was found to have moderate cerebellar atrophy on brain MRI, likely consistent with the attenuated clinical course reported in juvenile-onset NCL[36].

## Discussion

The standard approach in exome sequencing studies is to consider rare non-synonymous variants as disease candidates. This mainly includes missense, nonsense and canonical splice variants while other variant types are mostly ignored[6]. The ability to interrogate non-amino-acid changing variants within coding genes (either synonymous or intronic) is currently very limited, owing to several reasons: (1) the reduced evolutionary constraints in those regions[19], which makes it harder to infer their functionality, (2) the huge numbers of potential variants that arise from these regions, (3) the large sample sizes required to cover those regions with WGS[7] and (4) the lack of appropriate methods to account for 1 and 2. Thus, one of the missing building blocks in exome analyses is the lack of reliable methods to infer potential damaging effects, much less pathogenicity, of synonymous variants. This is becoming more of an issue whenWGS data are concerned due to the vast non-coding intronic variants that need to be assessed.

While existing methods are able to prioritize synonymous and intronic variants, they lack the specificity required for detection of causal variants. Several hundreds of sequenced whole genomes can produce millions of intronic variants and low specificity scores will leave researchers chasing hundreds of thousands of suspected variants. TraP was designed to handle exactly this issue. While we aimed at developing a successful predictor, maintaining the highest specificity was a top priority in TraP design. Discarding 99.4% of intronic variants as benign, while still maintaining a good evaluation of pathogenic variants, finally enables working with the vast numbers of variants arising from intronic regions.

TraP is not an ensemble of annotations but a simplistic model, taking into consideration mostly sequence and transcript information. Due to TraP being attentive to indirect splicing regulatory effects, it finds pathogenicity where scores searching for conservation, such as GERP++ and also CADD to an extent,

are blinded. In support of this statement, TraP identifies pathogenic variants that are not conserved, yet still have rare population frequencies. Doing this without prior population frequency information, and in contrast to conservation, suggests that TraP identifies pathogenic events that were not selected against during vertebrate evolution, but *are* selected against in human populations. This conclusion is supported by the fact that the highest complexity of alternative splicing is found in primates and by the species-specific nature of splicing regulation, that is directed mostly by *cis*-acting elements[21]. While identification of protein coding causal variants is made clearer because of their high evolutionary conservation, regulatory elements and highly skipped exons have much higher evolutionary turnover rate[37, 38], again, suggesting that conservation is a weaker marker for those regions.

While the above non-conserved and potentially pathogenic variants account for TraP's advantage over GERP++, the advantage over CADD might be due to the nature of ensemble annotation tools vs. more specifically aimed solutions. The CADD model incorporates many features, including conservation, epigenetic modifications and more. While the multitude of features is an advantage for the generic use of CADD and the ability to apply it over the whole genome, it is a disadvantage when applied to a region that may be weakly affected by many of these features. The focus of TraP on transcript affecting features renders it more useful when considering intronic and synonymous variants inside the gene region.

To facilitate usability among the general user community we hereby describe the three recommended thresholds based on the allele frequency tests that were introduced in the Results section (Fig. 2d). We consider a TraP score below 0.459 to, in general, be enriched for benign variants. We consider scores $\geq 0.459$ and $< 0.93$ as the intermediate pathogenic range, akin to possibly damaging classifications. These ranges are enriched for cryptic slice sites, effects of *cis*-acting regulatory sequences and weak to intermediate splice region changes. TraP scores $\geq 0.93$ are most likely to damage the final transcript and are considered as probably damaging. These variants are enriched for strong splice region changes and strong cryptic splice site creations.

Finally, we would like to note that by construction, TraP is specifically trained on pathogenic variants causing severe Mendelian disorders. It is therefore expected that TraP will perform optimally on severe and highly-penetrant Mendelian disorders. The application of the TraP scoring framework to more common and complex disorder risk allele predictions should be considered more carefully since TraP was not customized to catalogues of variants contributing to risk in these more genetically complex settings.

We conclude that TraP offers a substantial advance over available methods in the identification of intronic and synonymous variation that cause disease by affecting the constellation of transcripts that a gene produces. For this reason, TraP should find immediate application in a broad range of human genetics studies including diagnostic sequencing, which seeks to identify pathogenic mutations in large-scale gene discovery efforts using collapsing and related analyses.

## Methods

**Datasets**. Synonymous variants training-set: the list of synonymous pathogenic variants were obtained by combining previously reported variants taken from three sources: Chamary et al.[39], Buske et al.[40] and the OMIM online database[41] (accessed on May 2015). Upon merging these lists, we identified 93 variants including eight overlapping variants. Next, the 85 distinct variants were curated by going over the literature of each variant and validating that these previously reported variants were indeed linked to a rare disease and were unambiguously synonymous by ensuring the variant has no other non-synonymous annotation across any other

overlapping transcript at that variant site. We did not filter based on current reference cohort minor allele frequency of the reported pathogenic variant among databases like ExAC. As a result of this curation we were left with 75 pathogenic-assigned training variants (Supplementary Data 2).

The list of benign variants was obtained from control trios published by Iossifov et al.[22]. All 402 de novo synonymous variants resided within the consensus coding sequence (CCDS) and were identified from individuals not ascertained for any specific disorder. De novo variants were used because they represent novel variants in the gene pool and thus we can be assured that the model does not train against population variants, and thus (critically) does not train against population allele frequency (Supplementary Data 2).

ExAC synonymous variants: 1.46 million synonymous variants were downloaded from the ExAC database[9] along with their ExAC global allele frequencies. Variants were filtered based on having a 'synonymous-coding' impact using the Ensembl Variant Effect Prediction impact table[42]; any variant having another consequence of higher impact than synonymous-coding was removed from analysis.

Whole genome sequence intronic variants: 776 whole-genome controls available at the Institute for Genomic Medicine at Columbia University Medical Center were analyzed for SNP detection. We applied the following filters for mapping quality: (1) minimum read depth of 10, (2) phred scale probability that the alternative allele is incorrectly specified (QUAL) greater than or equal to 40, (3) mapping quality (MQ) greater than or equal to 30 and (4) quality by depth (QD) greater than or equal to 2. Variants were filtered based on having an 'intronic' impact using the Ensembl Variant Effect Prediction impact table[42]; any variant having another consequence of higher impact than 'intronic' was removed from analysis. The final intronic dataset consisted of 1.5 million variants that were randomly selected from 18,377,624 intronic variants that passed the above filters.

ClinVar pathogenic and benign datasets: 4,418 synonymous variants were obtained from the ClinVar database[23] that had a clinical significance of either 'pathogenic' or 'benign' and had CADD values. The clinical significance impact was decided in ClinVar based on the recommended rules by the ACMG[24]. Of the 4,418 variants, 66 marked as 'pathogenic' and 4,352 as 'benign'.

3,266 intronic variants, were obtained from ClinVar[23] that had a clinical significance of either 'pathogenic' or 'benign' and had CADD values. 452 are marked as 'pathogenic' by ACMG standards and 2,814 as 'benign'.

Epi4K dataset: The Epi4K consortium exome database consists of 281 epilepsy family trios[25, 26].103 de novo synonymous variants (Supplementary Data 6) were extracted based on the following criteria:

(1) The read depth in both parents and proband should be greater than or equal to 10, (2) phred scale probability that the alternative allele is incorrectly specified (QUAL) greater than 50, (3) genotype quality (GQ) greater than 20, (4) alternative allele quality by depth (QD) greater than 2, (5) mapping quality (MQ) greater than 40, (6) no reads in either parent or 1,631 internal controls should carry the alternate allele, (7) at least 20% of the reads in the child should carry the alternate allele, (8) at least three variant alleles must be observed in the proband, (9) quality control for the allele status in the EVS database[10] should not be missing or fail and (10) the alternative allele should be missing from the ExAC and EVS databases[9, 10].

**Model construction: information acquisition**. Each variant is first identified within the Genome Reference Consortium Human Genome build 37. Next, the following annotations are gathered: (a) All the transcripts that the variant resides in, or all the transcripts in the nearest gene, if gene ids are not provided. (b) All the exon/intron coordinates in which the variant resides, and (c) distances from both splice sites in each case (Supplementary Data 1).

The GERP++ Rejected Substitutions score is added as a measure of evolutionary conservation[15]. The GERP++ score measures evolutionary constraints acting on a specific coordinate and is the only external score not calculated by TraP.

**Model construction: feature extraction**. Once variant annotation information is gathered, all sequence effects that are caused by the variant are then calculated and the features are extracted. Below is a high-level description of the extracted features. The equations used for computing each of the 20 features used in the TraP model are also depicted (Supplementary Methods).

Splice site changes: any change to the splice site motif is calculated using a Position Specific Scoring Matrix (PSSM) based on all human exons. Splice site strength before the substitution is calculated for both the 3′ splice site (3′ss) and 5′ splice site (5′ss) of the harboring exon (or nearest exon in case variant is in an intron). Next, if the variant is within the splice site region, the splice site is scored again after the substitution. 3′ss is regarded as 20 nt upstream to the exon-intron junction and the first 3 nt of the exon. The 5′ss is regarded as the last 3 nt of the exon and the first 6 nt of the downstream intron.

Cryptic splice site creation/disruption: if the variant creates or disrupts a canonical sequence (AG/GT), the flanking sequence will be calculated for its similarity to a splice site motif. The cryptic splice site PSSM score will be calculated both with and without the variant.

Interactions between splice sites: differences between existing and new splice sites are also calculated. These differences are later used as splicing effect weight factors to calculate more complex features such as the Splice Site Overall Score and the Variant Splice Score (F11 and F20 in Supplementary Methods and

Supplementary Data 1). This follows the logic that an exon with a weak splice site will be highly affected by a variant creating a strong splice site, while a strong existing splice site will have no such effect.

Splicing regulatory binding sites: TraP construction pipeline loads four datasets of major splicing regulatory proteins: SRSF1[43], SRSF2[44], SRSF5[43] and SRSF6[43], and one set of splicing silencer sequences calculated in silico[45]. The variant is then tested for disruptions or creations of binding site sequences for any and all of the above regulatory sets.

CpG effects: DNA methylation changes can occur if the variant creates or disrupts a CpG di-nucleotide. Recent studies show that DNA-methylation affects the processes of transcription by changing the rate of RNA-polymerase II and also affect exon recognition, thus might contribute to the damaging effect of a variant[46]. This feature was eventually not incorporated into the final model since it did not add to the model's ability to distinguish pathogenic variants.

Overall, 14 general properties of the variant (such as coordinate, gene name, etc.) and 32 features are either collected in the information acquisition process or calculated by the feature extraction pipeline for each variant, of which 20 features are used in the TraP model (Supplementary Data 1).

Each of the 20 selected features' independent ability to differentiate between TraP-predicted pathogenic and benign variants in the ExAC 1.46 M synonymous variants dataset was also examined using frequency distributions (Supplementary Figs. 5–24). This was done separately for TraP-predicted pathogenic variants (TraP ≥ 0.459) and TraP-predicted benign variants (TraP < 0.459). As the values of the TraP features are not always normally distributed in the ExAC dataset, we used a non-parametric Mann–Whitney $U$-test to test the null hypothesis that the distribution of the values of each feature for TraP-predicted pathogenic variants is equal to the distribution for TraP-predicted benign variants (Supplementary Figs. 5–24, bottom line). 18 of the 20 TraP features have a significant difference (i.e., ability to discriminate) between high and low TraP variants ($P$-value < $2.2 \times 10^{-16}$). Of interest, the two remaining features are related to splicing regulatory functions: the 'combined ESR Score' and 'Negated ESR Score' features achieve a $P$-value of 0.06 and 0.09, respectively. This suggests that contribution to TraP originating from cis-acting elements of regulatory proteins are not as straightforward as that of the other features. We also provide a Spearman correlation matrix for the 20 features and the TraP score itself, using the same ExAC 1.46 M variants dataset, to help highlight the independent information that each feature provides to the TraP model (Supplementary Fig. 25).

**Model construction: random forest model training and cross-validation**. The random forest is an ensemble-learning algorithm designed to perform classification that uses a collection of decision trees, each tree is made of a random selection of features[47].

We used the randomForest package in R[48] to classify variants as either pathogenic (1) or benign (0). The model uses 1,000 decision trees, each with square ($N$) random set of features, $N$ being the overall number of features used. A probability is calculated as the fraction of tree 'votes' classifying a variant as damaging. Thus, a TraP score of 0.45 means that 450 trees considered a variant as pathogenic.

The model was trained using the entire variant dataset (75 pathogenic variants and 402 controls) and accuracy of 91.82% was evaluated based on the 8.18% out-of-bag estimation of error. As another way to assess the accuracy of the model, we also performed the following cross-validation scheme: 10% cross-validation (CV) was performed on the full dataset, each time building the model using 90% of the data and using it to score the rest 10%. When all samples were scored, an AUC was calculated using the true and false positive rates. The dataset was then shuffled and basic 10%-CV scheme was repeated again to calculate a second AUC. Dataset shuffling was repeated ten times and ten calculated AUCs were eventually averaged into an overall AUC of 0.86. The feature importance table was also calculated using the randomForest package and it corresponds to the mean decrease in Gini coefficient of each feature.

All scripts and tools were developed using the perl script language and the R statistical computing program[49], the latter was also used for both statistical analysis and graphical display.

**Computing a PSSM Score**. The algorithms used to score splice site PSSMs have been previously published[19]. In summary, to score the splice sites, it is necessary to obtain the genome's PSSMs of the 3′ and 5′ splice sites incorporating all human exons. The 3′ss was defined as the 20 intronic nucleotides upstream of exons and the first three exonic nucleotides, whereas the 5′ss was defined as the three terminal exonic nucleotides and the first six intronic nucleotides of the downstream intron. Only G[T/C]-AG exons were used for this analysis. Subsequently, every splice site was scored based on its adherence to the human PSSM, as follows:

$$\text{score} = \sum_{i=1}^{K} \log_2(f_{i,A_i})$$

Where $A$ is the sequence motif to be scored, $K$ is the motif length and $f_{i,A_i}$, is the PSSM frequency of character $A_i$ that is found in the $i$-th nucleotide in the motif, as in Shapiro and Senepathy[50]. Next, the score was normalized between 0 and 100 as

follows:

$$score = 100 \times (score - Min)/(Max - Min)$$

Where Min and Max are the minimum and maximum splice site scores overall motifs in that genome.

**Minigene construction.** Exon 10 and flanking intronic sequence was amplified from genomic DNA from the patient and parent using primers *MFSD8*-splice-XbaI-F: ATCATCTTCTAGATCTACTTTTTGTGTCCCAGAC and *MFSD8*-splice-Xho1-R3: GAGACTCGAGCAAAACCATTGCAGTGCATTA CTTGTTG. Forward and reverse primers introduced the Xba1 and Xho1 restriction sites used in standard cloning. The amplified mutant and WT products were digested and cloned into the pI-12 plasmid (pI-12 was a gift from Shelton Bradrick Ph.D and Mariano Garcia-Blanco Ph.D: Addgene # 24273[51]). Sanger sequencing confirmed insert of WT (minigene *MFSD8*-splice10-WT) and mutant (minigene MFSD8-splice10-mut) exon 10 and flanking intronic regions.

**Overexpression and splice isoform analysis.** HEK293T cells were transfected in triplicate with 1.5 μg of either mini gene and 0.5 μg of pCMV-AC-GFP vector to assess transfection efficiency. Total RNA was isolated 24 h after transfection and used for reverse transcription using Superscript III (ThermoFisher, 18080). The cDNA was amplified using the following primers to investigate effect of mutation on splicing (T7 promoter primer: TAATACGACTCACTATAGG and Sp6 promoter primer: ATTTAGGTGACACTATAGAA). PCR products were run on a 1% agarose gel. ImageJ software was used to quantify band intensity. Intensities were normalized to GAPDH.

**Data availability**. TraP is pre-computed for all human protein coding genes and is available at http://trap-score.org/

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

## Acknowledgements

We thank Dr. David Burstein for his support with model design. We acknowledge the Epilepsy Phenome/Genome Project and the Epi4K Consortium for use of the sequence data generated from epilepsy trios. Funding: National Institute of Neurological Disorders and Stroke (U01NS077303, U01NS053998, U01NS077274, U01NS077276, U01NS077303, U01NS077364, U01NS077367, and U01NS077275). We acknowledge the OCD Collaborative Genetics Association Study (OCGAS) for use of the sequence data generated from OCD trios. We acknowledge V. Shashi and the Duke sequencing clinic for contribution of the *MFSD8* trio case. We are grateful to the following individuals or groups for the contributions of control samples: D. Daskalakis; R Buckley; Y. Liu; N. Calakos; C. Chen; S. Raskin; B. Haynes; A. Holden; E. Behr; J. McEvoy; E. Holtzman; S. Kerns; D. Levy; H. Oster; D. Marchuk; J. Hoover-Fong, N. L. Sobreira and D. Valle; A. Poduri; Z. Farfel, D. Lancet, and E. Pras; R. Gbadegesin and M. Winn; Y. Jiang; S. Palmer; C. Depondt; V. Shashi; M. Carrington; ALS Sequencing Consortium; M. Harms, T. Miller, and A. Pestronk; R. Bedlack; B. Brown; N. Shneider; S. Gibson; J. Ravits; A. Gitler; J. Glass; F. Baas; S. Appel and E. Simpson; G. Rouleau; National Institute of Allergy and Infectious Diseases Center for HIV/AIDS Vaccine Immunology (CHAVI); National Institute of Allergy and Infectious Diseases Center for HIV/AIDS Vaccine Immunology and Immunogen Discovery; F. McMahon, N. Akula; K. Welsh-Bomer, C. Hulette, J. Burke; M. Connors, L. Morris, and the CHAVI investigators. The collection of control samples and data was funded in part by: Gilead Sciences, Inc.; Biogen, Inc.; B57 SAIC-Fredrick Inc M11-074; The Ellison Medical Foundation New Scholar award AG-NS-0441-08; National Institute of Mental Health (K01MH098126, R01MH097993; RC2MH089915); National Institute of Allergy and Infectious Diseases (1R56AI098588-01A1); National Human Genome Research Institute (U01HG007672); National Institutes of Health (U01MH105670); and National Institute of Allergy and Infectious Diseases Center (U19-AI067854); and the National Institute on Aging (P30AG028377).

## Author contributions

D.B.G. and S.G. designed the study. S.G. developed and trained the model and wrote the scripts required for calculating features and scoring. Q.W., S.P.,M.H., E.L.H., F.L.C and S.G. constructed datasets and annotations. S.G. performed analysis of all datasets. Z.R. developed the TraP database and website. K.M.M. carried out mini-gene construct experiments. K.S. facilitated clinical assessment. S.G. wrote the original draft. S.P., M.H., E.L.H., K.M.M., F.R., M.B. and D.B.G. performed review and editing. D.B.G supervised the study and acquired funding for the study.
