## [Peer Review file · Nature Communications]

Reviewers' comments:

Reviewer #1 (Remarks to the Author):

Gelfman et al. develop a score which uses sequence context alterations to predict the pathogenicity of synonymous and non-coding variation. The score differentiates itself from other methods such as GERP++ and CADD in that it doesn't focus on coding variation but instead on the class of genetic variation that usually is considered benign by default: synonymous/non-coding.

Overall, this is a nice study that appears to specialize in predicting pathogenicity of synonymous and non-coding variation. The method appears to perform better than CADD for synonymous/non-coding variation. This score seems to be useful as one could use established prediction scores such as CADD, fitCons to predict missense sites and then TraP to predict synonymous/non-coding sites.

Below are my comments, most of which are minor:

1) page 6: TraP Properties of Variants in the General Population - its mentioned that ExAC contains a small representation of other diseases. In fact, the majority of samples in ExAC are from disease studies including, T2D, MI, schizophrenia, cancer, and mendelian diseases.

2) page 6: based on the ExAC results, it says that only 2.16% of synonymous variants have high TraP scores and only 0.6% of intronic variants have high TraP scores. If these numbers are to be believe, then it indicates that >98% of synonymous/intronic variants are benign as expected. Perhaps this could be included more prominently in the abstract and/or discussion since I think its still unclear how often pathogenic synonymous/intronic variants occur.

3) page 8: TraP scores correlate with minor allele frequency, presumably because TraP can identify pathogenic variants that are under constraint due to natural selection. It says that synonymous variants with high TraP scores have lower allele frequencies than non-synonymous variants. This is presumably because the majority of non-synonymous variants are benign and therefore aren't under constraint. This appears to contradict results on page 7 which says that the "median GERP++ score for these intronic variants that are considered pathogenic by TraP is only 0.12, suggesting that half of them are not under evolutionary constraint. Aren't these two results conflicting with each other?

4) page 8 - correlate with minor allele frequency: it's mentioned that GERP++ is a main feature of TraP. There seems to be some circularity since GERP++ is a metric for evolutionary constraint, isn't it expected that TraP should correlate with minor allele frequency, since that is a feature of the TraP score? One would would like to see if the correlation with minor allele frequency still exists without inclusion of GERP++ as a feature.

5) page 10, identifying pathogenicity in a synonymous dataset: it appears TraP can identify true pathogenic variants with high specificity but moderate sensitivity The moderate sensitivity suggests that its mis-scoring a number of pathogenic variants to be benign. So presumably these are synonymous sites that cause pathogenicity apart from its effect on the mRNA transcript. Could it be that these ones affect through other mechanisms such as translation?

6) The application of TraP to a complex disease in epilepsy and a single case study for a Mendelian disease are nice proof-of-principle examples being applied to disease settings.

Reviewer #2 (Remarks to the Author):

The paper proposes a novel scoring method for non-coding genic variants, and a webserver of pre-computed scores is provided to the public.

The method is novel and relevant for researchers and clinical practitioners of genome sequencing alike.

The results section of the paper is clearly written and numerical results on various data sets are convincing.

The method is shown to work well in discriminating known pathogenic variants from others on several data sets, and scores have been shown to be anti-correlated with allele frequency.

Where the paper in its current form falls short is to provide a more detailed analysis of the Test set variants predicted to be pathogenic.

Questions that I would like to see answered include the following:

Some general summary statistics over these variants would be helpful.

What is it that drives predicted pathogenicity?

What were the model-features that drive high scores in the test data sets?

A statistical analysis comparing the distribution of test set variants with high and low scores would be interesting.

Where variants with high scores enriched in particular regions of the genome?

Are they located close to any of the 75 training variants?

Additional details and a more thorough description of the methods should be provided to improve clarity, reproducibility, and would help other method developers to build on the proposed method. Source code would be desirable.

The methods section reads handwavy throughout.

While you provide a feature importance in Table S3, it is unclear to me, how you computed this feature importance.

How has the PSSM be computed?

Table S1 has incomplete formulas. How have these formulas been derived?

What are the

While the list of pathogenic (and control) training variants is provided, additional description of how these had been selected would be helpful. How many variants were considered in total? Why have others not been selected?

Reviewer #1:

1) page 6: TraP Properties of Variants in the General Population - its mentioned that ExAC contains a small representation of other diseases. In fact, the majority of samples in ExAC are from disease studies including, T2D, MI, schizophrenia, cancer, and mendelian diseases.

This is an important point. The ExAC database represents a human population reference cohort that also includes samples from various common disorder studies such as type-2 diabetes, schizophrenia and myocardial infarction. ExAC has, however, made every effort to “remove individuals affected by severe pediatric disease, so this data set should serve as a useful reference set of allele frequencies for severe disease studies.” We now clarify this in our revised manuscript (Page 6; lines 11 – 16). We also include in the discussion that we expect TraP to perform optimally on severe and highly-penetrant Mendelian disorders. This is also in part by construction since TraP is specifically trained on pathogenic variants causing severe Mendelian disorders. Furthermore, the application of the TraP scoring framework to more common and complex disorder risk allele predictions should be considered more carefully since TraP was not customized to catalogues of variants contributing to risk in these more genetically complex settings (Page 19; lines 26 – 31).

2) page 6: based on the ExAC results, it says that only 2.16% of synonymous variants have high TraP scores and only 0.6% of intronic variants have high TraP scores. If these numbers are to be believe, then it indicates that >98% of synonymous/intronic variants are benign as expected. Perhaps this could be included more prominently in the abstract and/or discussion since I think its still unclear how often pathogenic synonymous/intronic variants occur.

This is an interesting suggestion. TraP currently classifies approximately 2.16% ExAC synonymous variants and 0.6% of control genomes intronic variants as pathogenic. We can expect that this small subset of ExAC pathogenic classified variants will be a mixture of clinically relevant variants that reside among ExAC population and also some background noise that can occur due to biological/technical reasons within the ExAC sample. We now include an important comment in the manuscript about how the ExAC reference cohort sample adopted in this study informs us that we can expect in any large human population sample the rate of TraP-based pathogenic-classified synonymous and intronic variants to be approximately 2% and 0.6%, respectively (Page 6; lines 23 – 28 and Page 8; lines 6 - 8). This is also an important practical point because this estimate allows users to understand what they can expect the background rate of TraP pathogenic-classified variants to be in any random sample; thus, giving users some indication of whether to be impressed by rates of TraP pathogenic-classified variants that they might identify among their disease-ascertained cohorts.

3) page 8: TraP scores correlate with minor allele frequency, presumably because TraP can identify pathogenic variants that are under constraint due to natural selection. It says that synonymous variants with high TraP scores have lower

allele frequencies than non-synonymous variants. This is presumably because the majority of non-synonymous variants are benign and therefore aren't under constraint. This appears to contradict results on page 7 which says that the "median GERP++ score for these intronic variants that are considered pathogenic by TraP is only 0.12, suggesting that half of them are not under evolutionary constraint. Aren't these two results conflicting with each other?

To answer this question, we initially focus on the variants that TraP predicts as pathogenic. Despite allele frequency not included in score construction, we found that the variants with a high TraP score preferentially have low minor allele frequencies, which is what we expect if those variants are and have been selected against in the human population. However, selection of this class of variants may not always show in evolutionary conservation because much of splicing regulation is species specific and is directed mostly by cis-acting elements¹. Thus, the regulation of splicing of a specific RNA sequence is not necessarily the same in other species. For this reason, a high TraP variant that has a strong regulatory function in human, does not necessarily have the same function in other species, and will therefore not always occur at highly conserved sites.

In contrast to non-coding variants, protein sequence is more conserved across species. In fact, according to the GERP++ study², the average GERP++ score of the coding sequence is > 2 , marking very strong conservation compared to the measured 0.12 GERP of intronic variants.

When considered as a class of variants, missense variants more frequently have higher consequences on protein-products, which explains why the coding sequence is often under stronger evolutionary constraint. As a result, we see that in general missense variants have lower allele frequencies compared to synonymous variants (figure 2D in the revised manuscript). The fact that we observe similar MAF between high TraP synonymous variants as missense variants is of great importance, as this observation suggests that this sub-group of synonymous variants that have high TraP scores do appear to be under similar levels of negative selection as the missense class of variants and might be as damaging. We have revised the relevant text in the revised manuscript (Page 9; line 27 – Page 10; line 2 and Page 18; line 16).

4) page 8 - correlate with minor allele frequency: it's mentioned that GERP++ is a main feature of TraP. There seems to be some circularity since GERP++ is a metric for evolutionary constraint, isn't it expected that TraP should correlate with minor allele frequency, since that is a feature of the TraP score? One would like to see if the correlation with minor allele frequency still exists without inclusion of GERP++ as a feature.

This is an important point and one we are glad to be able to clarify in our revision. In our revised version, we have included two additional analyses to better illustrate the nature of the relationships between TraP and GERP++ in the context of correlations with MAF.

In addition to the general negative correlation observed with MAF for all variants, we specifically examine the "Cryptic 5' Splice Site Score" (feature F7) class of variants as

this feature displays high contribution to the variants TraP considers pathogenic (through the model importance table, supplementary table S3). We focused the MAF correlation assessments on this class of variants as they represent a class of variants expected to be particularly sensitive to TraP but not necessarily to conservation.

Taking the TraP and GERP++ score 5% intervals for the ExAC dataset of 1.46M synonymous variants, we find that the correlation of GERP++ with MAF is -0.82 (p-value = 4.7×10^{-06}) and TraP is -0.52 (p-value = 0.021). Demonstrating that genome-wide GERP++ has a stronger correlation with MAF than TraP. However, focusing on the subset of 29,985 synonymous variants that are classed as creating a new, strong cryptic 5'ss, we find that the TraP correlation with MAF for this class of variants is stronger $r = -0.84$ (p-value $< 2.2 \times 10^{-16}$) compared to the correlation between the variant sites conservation (GERP++) and MAF, $r = -0.59$ (p-value = 0.0069, Figure 2B and 2C in the revised manuscript). When Sub-selecting cryptic splice site variants that are predicted as pathogenic (TraP ≥ 0.459 , 6,328 variants), TraP correlation is -0.51 (p-value = 0.025) while GERP++ correlation is -0.15 (p-value = 0.53, new supplementary figure S4). These results clearly exhibit how TraP identifies splice potential variants that are deleterious (thus, strongly selected against in the human lineage), but do not necessarily have a strong conservation signature. We have clarified these results in the revised manuscript (Page 8; line 26 – Page 9; line 11, Figure 2B-2C and supplementary figure S4).

We also constructed an additional TraP model without GERP++ included as a feature and tested it using the MAF distribution among the ExAC synonymous variants. In comparison to the original model which achieved an accuracy of 91.8%, the model without GERP++ as a feature achieved an accuracy of 90.5%. Comparing allele frequencies of high and low TraP variants in the GERP-less model again shows a similar strong decrease in MAF distribution as seen with the original TraP, but as expected this modification in the score requires adjusted TraP thresholds (see Figure 1 in this response, which is also added as a new Figure 2E). We observe a significant decrease in MAF distribution when comparing all synonymous variants to the subset of synonymous variants with high GERP-less TraP scores (Figure 1 in this response file, yellow and orange bars, MW-test, p-value = 2.01×10^{-08}). In this analysis, a TraP threshold of 0.66 was used to classify high TraP scores, which corresponds to the 25% percentile of the GERP-less TraP scores. When further increasing the threshold (TraP > 0.955 , i.e., the 75% percentile among the GERP-less TraP training set), the MAF distribution further decreases, and is even significantly below the MAF of synonymous variants with TraP > 0.66 and < 0.955 (orange and red bars, MW-test, p-value = 2.39×10^{-12}). These results highlight that the GERP-less TraP captures the relevant signals among possible variants to help identify this class of variants that damage the transcript and are selected against in the human population. Last, in our previous text we found an inaccuracy in the reported TraP thresholds for similar MAF with NS (0.835 instead of 0.459) and lower than NS (0.93 instead of 0.835). We have updated the relevant section in the revised manuscript accordingly (Page 10; line 4 - 23 and figure 2E).

To facilitate usability among the more general user community we also specifically describe the three recommended thresholds. We consider a TraP score below 0.459 to in general be enriched for benign variants. We consider scores ≥ 0.459 and < 0.93 as the intermediate pathogenic range, akin to possibly damaging classifications. These ranges are enriched for strong cryptic splice site, strong effects of cis-acting regulatory sequences and weak to intermediate splice region changes. TraP scores of ≥ 0.93 are most likely to damage the final transcript and are considered as probably damaging. These are enriched for strong splice region changes and very strong cryptic splice site creations. We now describe this in the revised manuscript (Page 19; lines 17 - 25).

Figure 1

5) page 10, identifying pathogenicity in a synonymous dataset: it appears TraP can identify true pathogenic variants with high specificity but moderate sensitivity suggests that its mis-scoring a number of pathogenic variants to be benign. So presumably these are synonymous sites that cause pathogenicity apart from its effect on the mRNA transcript. Could it be that these ones affect through other mechanisms such as translation?

This is a great point. Other biological mechanisms can disturb protein function and expression and these may not be part of the signal detected by TraP. TraP, by construction, focuses on identifying transcript effects that can be strong protein loss or gain of function, yet variants might also have less prominent effects on translation and expression. Some options that might affect gene expression include changes to the RNA secondary structure that will affect translation efficiency, while another effect might be a codon usage bias of synonymous variation. While TraP in its current form is doing very well in predicting pathogenicity, incorporating these additional disease mechanisms is something of interest for future developments.

6) The application of TraP to a complex disease in epilepsy and a single case study for a Mendelian disease are nice proof-of-principle examples being applied to disease settings.

We thank the reviewer for this comment.

Reviewer #2:

1) Where the paper in its current form falls short is to provide a more detailed analysis of the Test set variants predicted to be pathogenic. Questions that I would like to see answered include the following: Some general summary statistics over these variants would be helpful.

We have revised the manuscript to include the relevant information regarding the test dataset and how the specific features affect the overall score of the variants. We have also included new analyses in the revised version that examine the behavior of each individual feature using the test dataset. To achieve this, we plotted the frequency distributions for each of the 20 features using the 1.46M ExAC synonymous variants. This was done separately for variants that are equal to or above the recommended pathogenic threshold ($\text{TraP} \geq 0.459$) and the variants below the threshold (benign; $\text{TraP} < 0.459$). All 20 figures, along with their corresponding Mann Whitney U tests, are now added as figures S6 - S25. In summary, we found that 18 of the 20 features have a significant difference (i.e., ability to discriminate) between high and low TraP variants ($p\text{-value} < 2.2 \times 10^{-16}$). Of interest, the two remaining features are related to splicing regulatory functions: the 'combined ESR Score' and 'Negated ESR Score' features achieve a $p\text{-value}$ of 0.06 and 0.09, respectively. This suggests that contribution to TraP originating from cis-acting elements of regulatory proteins are not as straightforward as the other features. We also provide a spearman correlation matrix for these twenty features and the TraP score, to help highlight the independent information that each provides to the TraP model (Figure S26).

2) What is it that drives predicted pathogenicity? What were the model-features that drive high scores in the test data sets?

This is an important question and we've revised our discussion to include some commentary on the individual features found to have a high contribution to the TraP score. As per our above response, we now include a supplemental figure for each of the 20 features to show the distribution of the feature scores when comparing the test variants with TraP scores < 0.459 (i.e., TraP benign predictions) versus those with TraP scores ≥ 0.459 (i.e., TraP pathogenic predictions). We have also separately included as an additional supplemental figure a spearman correlation matrix between features and between each feature and the final TraP score. In general, we derive feature contributions from the importance table in supplementary figure S3, that is further described in answer to comment #6 below. While there is no specific feature that has an exceptionally high importance, we find that the features that strongly contribute to the final TraP pathogenicity predictions are mostly features that are designed to capture interactions between various variant attributes. The strongest contributor is the Variant

Splice Score (F20 in answer to comment #8) that incorporates original and cryptic splice site changes as well as regulatory changes and conservation. The 2nd contributing feature is the conservation score, followed by the Variant Regulatory Score (F19) that is similar to F20 but not incorporating splice site specific effects in its calculation. Also important are the Splice Site Overall Score (F11) that is made of both 3' and 5' splice site negative effects and the Cryptic 5'ss Score (F7), that points to the stronger contributions of variants that create new 5' splice sites. We now also comment on these features in the results of the revised manuscript (Page 5; lines 10 – 21).

3) A statistical analysis comparing the distribution of test set variants with high and low scores would be interesting.

We believe this has been addressed in response to comments #1 and #2, above.

4) Were variants with high scores enriched in particular regions of the genome? Are they located close to any of the 75 training variants?

This is an interesting question and one we had not previously considered to explore. We have now plotted the distribution of variants considered as pathogenic ($\text{TraP} \geq 0.459$) and benign ($\text{TraP} < 0.459$) per chromosome (figure 2 below). As can be seen, there is no clear difference in distributions of the pathogenic and benign variants along the various chromosomes and they both closely correspond with the tally of genes in a chromosome. This figure is added to the revised manuscript as a new supplementary figure S1 (as also Page 6; line 29 – Page 7; line 4).

We have also constructed an analysis to examine the distances between the training set pathogenic variants and the test set predicted pathogenic variants. Considering the local sequence content flanking pathogenic-assigned training variants, we find that within 100 flanking bases and subsequently within 1,000 and 10,000 flanking bases the percentage of TraP pathogenic-predicted variants are $0.012\% \pm 0.033$, $0.035\% \pm 0.057$ and $0.15\% \pm 0.23$, respectively. We then did similar for 1,000 randomly selected TraP-predicted pathogenic synonymous variants from the 1.46M ExAC synonymous dataset. We found that the neighboring 100 flanking bases and subsequently 1,000 and 10,000 bases had comparable rates of TraP-predicted pathogenicity of $0.012\% \pm 0.034$, $0.035\% \pm 0.069$ and $0.13\% \pm 0.19$, respectively. All three distance groups (100bp, 1,000bp and 10,000bp) are not statistically different between the training set and the randomly selected TraP-predicted pathogenic variants. These results show that the 75 pathogenic-assigned training variants are achieving similar local rates of TraP pathogenicity scores as pathogenic-predicted variants that were not part of the training set. We find that this specific assessment ensures the robustness of the predictions and we now include it in the revised manuscript (Page 7; lines 5 – 19).

Figure 2

5) Additional details and a more thorough description of the methods should be provided to improve clarity, reproducibility, and would help other method developers to build on the proposed method. Source code would be desirable. The methods section reads handwavy throughout.

This is an important point - we have re-written the methods section and provided the necessary information to make the algorithms and process reproducible. We also list all equations used to generate each of the features as elaborated below in response to comment 8). We have also been advised that due to current licensing efforts of this innovative tool, it would not be appropriate to publicly share the source code at this time. We therefore have ensured that all 3.9 billion calculated TraP scores are publicly accessible through the TraP web-server (<http://trap-score.org>) – this will enable researchers and clinical labs to explore TraP scores within their sequence data and make immediate use of it.

6) While you provide a feature importance in Table S3, it is unclear to me, how you computed this feature importance.

The feature importance table was computed using the function *importance* from the randomForest R package. The function extracts the importance information that corresponds to the mean decrease in Gini coefficient and is a measure of how each

variable contributes to the homogeneity of the nodes in the resulting model. A higher importance, or higher decrease in Gini index, means that a particular feature plays a greater role in partitioning the data into the defined classes of benign and pathogenic. The use of the mean decrease in Gini index as the importance measurement is now added to the revised manuscript (Page 25; lines 13 - 14).

7) How has the PSSM be computed?

The algorithms we used to score splice site PSSMs have been previously published³ and we now emphasize the link to the PSSM formulation in our resubmission (Page 25; line 19 – Page 26; line 7). In summary, to score the splice sites, it is necessary to obtain the genomes Position Specific Scoring Matrices (PSSMs) of the 3' and 5' splice sites incorporating all human exons. The 3'ss was defined as the 20 intronic nucleotides upstream of exons and the first three exonic nucleotides, whereas the 5'ss was defined as the three terminal exonic nucleotides and the first six intronic nucleotides of the downstream intron. Only G[T/C]-AG exons were used for this analysis. Subsequently, every splice site was scored based on its adherence to the human PSSM as follows:

$$score = \sum_{i=1}^K \log_2(f_{i,A_i})$$

Where A is the sequence motif to be scored, K is the motif length and f_{i,A_i} is the PSSM frequency at character A_i that is found in the i -th nucleotide in the motif, as in Shapiro and Senepathy⁴. Next, the score was normalized between 0 and 100 as follows:

$$score = 100 \times (score - \text{Min}) / (\text{Max} - \text{Min})$$

Where Min and Max are the minimum and maximum splice site scores over all motifs in that genome. In addition to the citation of the original work we have now also added this information to the Methods section in the revised manuscript.

8) Table S1 has incomplete formulas. How have these formulas been derived?

We have now substantially revised these sections to include technical details for the relevant features, beyond the high-level concept that each feature captures. In the revised manuscript, each feature now has a designated paragraph. For review convenience, we have also included below the full explanation for each feature as it has been added to the manuscript as supplementary Methods.

F1: Variant Status Factor. 1- the variant is either only within exons or only within introns, 2 – the variant is in an alternatively spliced exon.

F2: Number of Affected Transcripts. The number of protein-coding transcripts that harbor this variant.

F3: Within 5'ss Region. The variant is within the 5'ss region, i.e: last 3 exonic bp and first 6 intronic bp.

F4: Within 3'ss Region. The variant is within the 3'ss region, i.e: last 20 intronic bp and first 3 exonic bp.

F5: Number of ESR and SS Enhancer Cases. A count of all cases that might positively affect inclusion: creation of enhancing SRPs sites, disruption of silencing SRPs sites and increased strength of original splice sites.

F6: Cryptic 3'ss score w/ variant. PSSM score computed around the variant if a new AG di-nucleotide is created.

F7: Cryptic 5'ss score w/ variant. PSSM score computed around the variant if a new GT di-nucleotide is created.

F8: GERP++ RS. Conservation score as obtained from the hg19 GERP database for the position in question.

The twelve features calculated below are designed to capture underlying interactions between non-independent attributes that affect the transcript synergistically rather than as single features. An example to this would be a variant that is creating a new strong cryptic splice site, the effect of such a variant might be different depending on the score of the exons' original splice site. The original splice site is considered as a variant attribute, and therefore this interaction between new and original splice site is an interaction between the variant's attributes. Using only the score of the splice site (both new and original) proved too weak when constructing the model, yet the interactions strengthened the model significantly. The following section explains the equations and rationale behind these more complex features:

F9 and F10: 3'ss and 5'ss Silencer Scores. The sum of difference across all transcripts between the reference splice site and the alternative splice site when the variant is within the splice site region and the new splice site is stronger than the original splice site. The rationale here is that the effect will correlate with the difference between splice sites and not their absolute strength.

$$F9 = \sum_{i=1}^{tmax} (Alt3_i - Ref3_i)$$

$$F10 = \sum_{i=1}^{tmax} (Alt5_i - Ref5_i)$$

Where $tmax$ is the number of transcripts for a specific gene, $Ref3$ and $Ref5$ are the PSSM scores for the reference 3' and 5' splice sites in the i -th transcript respectively,

$Alt3$ and $Alt5$ are the PSSM scores for the alternative 3' and 5' splice sites in the i -th transcript. F_9 and F_{10} are the silencing scores for each splice site.

F11: Splice Site Overall score. The sum of F_9 and F_{10} as the total change in splice sites across all transcripts.

F12: Silencing Effect Score: Overall difference between reference splice sites and a new splice site combining the information from both splice sites across all transcripts. Here we calculate the interaction between the overall difference of new stronger splice site and the original splice site across all the transcripts of the gene, while taking into account normalized PSSM scores. The rationale behind this formula is that a variant will affect all transcripts of a gene and will be dependent in the relative strength of the strongest splice site.

$$min3 = 22.995$$

$$min5 = 32.083$$

$$norm3 = \frac{\max(Ref3, Alt3)}{min3}$$

$$norm5 = \frac{\max(Ref5, Alt5)}{min5}$$

$$F12 = \sum_{i=1}^{tmax} (norm3 \times (Alt3_i - Ref3_i) + norm5 \times (Alt5_i - Ref5_i))$$

$$TES = \sum_{i=1}^{tmax} (norm3 \times \text{abs}(Ref3_i - Alt3_i) + norm5 \times \text{abs}(Ref5_i - Alt5_i))$$

Where $tmax$ is the number of transcripts for the specific gene, $min3$ and $min5$ are the PSSM scores for the 3' and 5' splice sites without a canonical splice site (used to weight the scores according to fold change from minimum score), $Ref3$ and $Ref5$ are the PSSM scores for the reference 3' and 5' splice sites, $Alt3$ and $Alt5$ are the PSSM scores for the alternative 3' and 5' splice sites, $F12$ is the silencing effect score and TES is a total effect score used later in the construction of the Variant Splice Score (F20).

F13 and F14: Cryptic 3'ss and 5'ss Effect Scores: these features calculate a score for a new cryptic splice site. The score is made of the difference between the newly created splice site PSSM and the reference sequence at that position.

$$F13 = c3alt - c3ref$$

$$F14 = c5alt - c5ref$$

Where $c3Alt$ and $c5Alt$ are the cryptic PSSM scores calculated for the variant, $c3Ref$ and $c5Ref$ are the PSSM scores calculated for the reference sequence, F13 and F14 are the final effect scores.

F15 and F16: Cryptic 3'ss and 5'ss Enhancer Scores. Similar to 3'ss and 5'ss Cryptic Effects (F13-F14), but with only positive differences examined, thus setting a zero value to features of variants that create cryptic splice sites that are weaker than the reference.

F17 and F18: Splicing regulatory binding site scores. TraP uses four datasets of major splicing regulatory proteins: SRSF1⁵, SRSF2⁶, SRSF5⁵ and SRSF6⁵, and one set of splicing silencer sequences calculated in-silico⁷. For each of these proteins, TraP is using a normalized rank score (between 0 and 1) indicating the ranking of a given sequence relative to the other sequences identified for that protein. The ranked datasets were obtained from Schwartz et al.⁸ and correspond to the sequences' p-values or PSSM log-odd scores. For each variant, TraP identifies creations or disruptions of sequences and uses the rank of the sequence to calculate the related feature. Thus, a creation of an enhancer sequence that ranks 0.9 will add +0.9 to an ESR enhancer score, while a creation of a silencer sequence that ranks 0.5 will add +0.5 to an ESR silencer score. Eventually, the ESR enhancer score will be the addition of all the creations of enhancers and disruptions of silencers for the five datasets. The ESR silencer score, accordingly, will be creations of silencers and disruptions of enhancers.

$$eeS = \left(\sum_{i=1}^{EGroups} \prod_{j=1}^{ESRs} VR_{ij} \right) + \left(\sum_{i=1}^{SGroups} \prod_{j=1}^{ESRs} RR_{ij} \right)$$

$$esS = \left(\sum_{i=1}^{EGroups} \prod_{j=1}^{ESRs} RR_{ij} \right) + \left(\sum_{i=1}^{SGroups} \prod_{j=1}^{ESRs} VR_{ij} \right)$$

$$F17 = esS - eeS$$

$$F18 = esS + eeS$$

Where EGroups are groups of Enhancer SRPs, ESRs are the sequences in each SRP datasets, RR is the rank of a reference sequence that was disrupted, VR is the rank of a sequence created by the variant, eeS is the ESR enhancer score and esS is the ESR silencer score. F17 is the negated ESR score, that is the overall tendency of the variant to either silencing or enhancing. F18 is the combined ESR score that holds the overall disturbance of regulatory sequences caused by the variant.

F19: Variant Regulatory Score. A feature that adds together all the effects that do not directly change the splice site region. Therefore, the following equations are using variant attributes calculated previously in

$$minCanonical3 = 44.836$$

$$\text{minCanonical5} = 62.556$$

$$\text{cryp} = \left(F13 \frac{\text{c3ss}}{100 - \text{minCanonical3}} \right) + \left(F14 \frac{\text{c5ss}}{100 - \text{minCanonical5}} \right)$$

$$\text{esr} = \text{esS} * \text{sCount} + \text{eeS} * \text{eCount}$$

$$F19 = \text{esr} \times F1 + F8 + \text{cryp}$$

Where *c3ss* and *c5ss* are cryptic splice sites scores (features F6 and F7), *c3ES* and *c5ES* are the cryptic effect scores (features F13 and F14), *cryp* is the calculated effect of cryptic splice site creation, *esS* and *eeS* are ESR enhancer and silencer scores (calculated for features F17 and F18), *sCount* is the number of silencing events, *eCount* is the number of enhancing events, *esr* is the total effect of ESR attributes, *F1* is the Variant Status Factor, *F8* is the GERP RS conservation score and *VRS* is the final Variant Regulatory Score.

F20: Variant Splice Score: the combined effects and interactions of regulatory and splice region features. The subtraction of the Variant Status Factor is acting to reduce the effect of a variant that is already inside an alternatively spliced exon.

$$\text{splice} = F9 + F10 + F12 + \text{TES} - F1$$

$$F20 = \text{esr} + F8 + \text{cryp} + \left(\frac{F13}{\text{minCanonical3}} \right) + \left(\frac{F14}{\text{minCanonical5}} \right) + \text{splice}$$

Where *F9* and *F10* are the silencer scores, *F12* and *TES* are the silencing and total effect scores, *splice* is the total addition of splice region effects, *esr* is the total effect of ESR attributes (calculated for F19), *F8* is the GERP RS conservation score, *cryp* is the effect of cryptic splice site creation (calculated for F19), *F13* and *F14* are the cryptic effect scores and *F20* is the final Variant Splice Score that incorporates attributes for splice junction, ESRs, conservation and cryptic splice sites.

9) While the list of pathogenic (and control) training variants is provided, additional description of how these had been selected would be helpful. How many variants were considered in total? Why have others not been selected?

We have included and updated the relevant text (Page 20; line 18 – Page 21; line 3). In brief, the list of synonymous variants were obtained by combining previously reported variants taken from three sources: Chamary et al.⁹, Buske et al.¹⁰ and the OMIM online database¹¹ (accessed on May 2015). Upon merging these lists, we identified 93 variants including eight overlapping variants. Next, the 85 distinct variants were curated by going over the literature of each variant and validating that these previously reported variants were indeed linked to a rare disease and were unambiguously synonymous by ensuring

the variant has no other non-synonymous annotation across any other overlapping transcript at that variant site. We did not filter based on current reference cohort minor allele frequency of the reported pathogenic variant among databases like ExAC. As a result of this curation we were left with 75 pathogenic-assigned training variants. We have also built the TraP model with the additional 66 un-curated synonymous variants obtained from ClinVar (used as a test set for figure 3A) as an alternative training set with a total of 141 pathogenic-assigned training variants. However, the model with the un-curated variants achieved an accuracy of 88.32%, considerably lower than the 91.8% of the model with the curated 75 pathogenic variants. This decreased accuracy was due to a decrease in model specificity and since TraP was constructed with an aim for highest specificity, this secondary model was not further evaluated.

The list of benign variants was obtained from control trios published in Iossifov et al.¹². All 402 *de novo* synonymous variants resided within the consensus coding sequence (CCDS) and were identified from individuals not ascertained for any specific disorder. *De novo* variants were used because they represent novel variants in the gene pool and thus we can be assured that the model does not train against population variants, and thus (critically) does not train against population allele frequency. We have not considered using a different dataset for these purposes.

References

1. Barbosa-Morais, N.L. *et al.* The evolutionary landscape of alternative splicing in vertebrate species. *Science* **338**, 1587-93 (2012).
2. Davydov, E.V. *et al.* Identifying a high fraction of the human genome to be under selective constraint using GERP++. *PLoS Comput Biol* **6**, e1001025 (2010).
3. Gelfman, S. *et al.* Changes in exon-intron structure during vertebrate evolution affect the splicing pattern of exons. *Genome Res* **22**, 35-50 (2012).
4. Shapiro, M.B. & Senapathy, P. RNA splice junctions of different classes of eukaryotes: sequence statistics and functional implications in gene expression. *Nucleic Acids Res* **15**, 7155-74 (1987).
5. Cartegni, L., Wang, J., Zhu, Z., Zhang, M.Q. & Krainer, A.R. ESEfinder: A web resource to identify exonic splicing enhancers. *Nucleic Acids Res* **31**, 3568-71 (2003).
6. Liu, H.X., Chew, S.L., Cartegni, L., Zhang, M.Q. & Krainer, A.R. Exonic splicing enhancer motif recognized by human SC35 under splicing conditions. *Mol Cell Biol* **20**, 1063-71 (2000).
7. Zhang, X.H. & Chasin, L.A. Computational definition of sequence motifs governing constitutive exon splicing. *Genes Dev* **18**, 1241-50 (2004).
8. Schwartz, S., Hall, E. & Ast, G. SROOGLE: webserver for integrative, user-friendly visualization of splicing signals. *Nucleic Acids Res* **37**, W189-92 (2009).
9. Chamary, J.V., Parmley, J.L. & Hurst, L.D. Hearing silence: non-neutral evolution at synonymous sites in mammals. *Nat Rev Genet* **7**, 98-108 (2006).
10. Buske, O.J., Manickaraj, A., Mital, S., Ray, P.N. & Brudno, M. Identification of deleterious synonymous variants in human genomes. *Bioinformatics* **29**, 1843-50 (2013).

11. Hamosh, A., Scott, A.F., Amberger, J.S., Bocchini, C.A. & McKusick, V.A. Online Mendelian Inheritance in Man (OMIM), a knowledgebase of human genes and genetic disorders. *Nucleic Acids Res* **33**, D514-7 (2005).
12. Iossifov, I. *et al.* De novo gene disruptions in children on the autistic spectrum. *Neuron* **74**, 285-99 (2012).

Reviewers' comments:

Reviewer #1 (Remarks to the Author):

The authors have adequately addressed most of my comments. Just a couple minor thoughts:

1. In the Response, new thresholds for intermediate pathogenic and probably damaging were determined, presumably for easier interpretation of the TRAP scores. Please include how these specific thresholds were defined.

2. Will the software to calculate TRAP scores be released publicly? Ideally, it would be a command line script that takes as input a single VCF file from WGS data and outputs TRAP scores.

Reviewer #2 (Remarks to the Author):

The paper has been improved in terms of the details provided.

The various results on MAF vs. scores provide strong evidence that the method seems to reliably predicts pathogenicity.

However, the added information should be better integrated/discussed in the text and Supplemental Figures should have informative captions.

Statistics and assumptions should be better motivated/explained (e.g. use of MWU tests).

Also, the fact that no source code is provided, makes the approach makes it harder to build on and compare to the results provided in the current method.

Detailed comments:

"... serve as a useful reference set of allele frequencies for severe disease studies."

Probably control set may be a better phrase than reference set in this case.

background noise and biological/technical reasons are mentioned for 2.16% pathogenicly classified synonymous variants. How about the possibility of False Positives? This should be mentioned/discussed.

Figure S1 only looks at aggregates per chromosome to compare distribution of benign vs. pathogenic.

This analysis is much too coarse to be informative, in particular to map enrichment of variants predicted to be pathogenic on the genome. Chromosomes are so large, that it would not be expected to find significant differences between pathogenic/non-pathogenic frequencies.

p 10 line 272:

identifying variants that (likely/presumably) are under stronger negative selection

As this statement is not further validated, this should be weakened.

line 272:

The word "fact" should be removed or evidence should be provided.

Figure 2BC: What do the dots in these plots show? How is data aggregated? The caption says there are 29k variants. The plots only show ~20 dots. This should be explained in the caption.

Figure 2DEF: Why are all the trap scores changed from panel to panel? The choices should be explained further in the legend. Coloring of bars also is confusing, as colors don't strictly correspond between panels.

Your responses to answer my questions on evaluating variants should be discussed in the text/Supplemental text. Bare Figures in the Supplement are not useful for the reader.

SI: Supplemental Figures should have self-contained captions (I could not find any) and should be discussed somewhere in the text (either main or SI).

What is being tested in the Mann Whitney U tests? The hypothesis should be formulated.

F1-F20 + Figures S6-S25: The scores should be better explained. What do the X-scales in each figure correspond to?

Figure S26: in the response you mention that this is a spearman correlation matrix. This should be mentioned in the caption. Also, over which set is this correlation computed?

Figure S6: What is the difference between 1 and 2?

Figure S8, S9: X-axis should probably be True/False, instead of 0/1.

Reviewer #1:

The authors have adequately addressed most of my comments. Just a couple minor thoughts:

- 1) In the Response, new thresholds for intermediate pathogenic and probably damaging were determined, presumably for easier interpretation of the TRAP scores. Please include how these specific thresholds were defined.**

These thresholds were defined based on the allele frequency tests presented in figure 2D and described on page 9; lines 4 – 25. In brief, a TraP threshold of 0.8 and above identifies synonymous variants with similar MAF distribution to non-synonymous (NS) variants. A TraP threshold of 0.93 and above (5,402 variants) identifies variants with significantly lower allele frequencies than the NS variant class (Figure 2D, red and bright blue columns, MW-test, p-value = 0.046).

- 2) Will the software to calculate TRAP scores be released publicly? Ideally, it would be a command line script that takes as input a single VCF file from WGS data and outputs TRAP scores.**

We have pre-computed the genome-wide scores, which are publicly available for individual variant or batch browsing and download at <http://trap-score.org> (Page 4; lines 6 – 7 and Page 20; lines 6 -7).

Reviewer #2:

The paper has been improved in terms of the details provided.

The various results on MAF vs. scores provide strong evidence that the method seems to reliably predicts pathogenicity.

- 1) However, the added information should be better integrated/discussed in the text and Supplemental Figures should have informative captions.**

We have added the relevant information to the supplementary text and figures as detailed further in answer to comments #2, #6 and #11 - #17 below.

2) Statistics and assumptions should be better motivated/explained (e.g. use of MWU tests).

The values of the TraP component features are not always normally distributed in the ExAC 1.46 synonymous variants dataset. Thus, we used a non-parametric Mann Whitney U test across all features to test the null hypothesis that the distribution of the values of each feature for TraP-predicted pathogenic variants is equal to the distribution for TraP-predicted benign variants. We describe the motivation for using the MWU tests in the revised text (Page 24; lines 10 - 14).

3) Also, the fact that no source code is provided, makes the approach makes it harder to build on and compare to the results provided in the current method.

We acknowledge the reviewer's interests to 'build on' the existing tool, which we would welcome through collaboration. As per response to reviewer #1, all possible 3.9 billion substitutions covering all human genes are pre-computed and their TraP-scores are publicly available via our TraP web-server (<http://trap-score.org>). This enables additional independent evaluations of the score.

In regards to the sharing of functional code, we have discussed this further with the Columbia Technology Ventures (CTV) team who has advised that we first request each reviewer confirms their agreement to the terms suggested by the CTV's legal advisor by using the following language:

"The authors are providing a copy of the TraP source code in reply to the reviewer's request. Please see the copy of the TraP source code attached. The authors note that the TraP source code and the TraP scores generated and presented by the TraP source code are protected by copyright. The authors grant the reviewer permission to make a single reproduction of the source code solely for the purpose of reviewing the manuscript, subject to the reviewer's agreement to destroy the reproduction upon completion of the review. The authors reserve all other rights under copyright."

4) "... serve as a useful reference set of allele frequencies for severe disease studies." Probably control set may be a better phrase than reference set in this case.

We have revised the text accordingly (Page 6; line 15).

5) background noise and biological/technical reasons are mentioned for 2.16% pathogenicly classified synonymous variants. How about the possibility of False Positives? This should be mentioned/discussed.

We have clarified this point accordingly (Page 6; line 25).

- 6) Figure S1 only looks at aggregates per chromosome to compare distribution of benign vs. pathogenic. This analysis is much too coarse to be informative, in particular to map enrichment of variants predicted to be pathogenic on the genome. Chromosomes are so large, that it would not be expected to find significant differences between pathogenic/non-pathogenic frequencies.**

This is a good point. We agree with the reviewer that the comprehensive analysis showing that the 2.16% TraP-predicted pathogenic variants are not preferentially located near the training set pathogenic variants sufficiently addresses this point (Page 6; line 29 - Page 7; line 12). As a result, we have removed the previous figure S1 and its related text from the revised manuscript.

- 7) p 10 line 272: identifying variants that (likely/presumably) are under stronger negative selection.
As this statement is not further validated, this should be weakened.**

We have revised the text accordingly (Page 9; line 25).

- 8) line 272:
The word "fact" should be removed or evidence should be provided.**

We have revised the text accordingly (Page 9; lines 27 - 28).

- 9) Figure 2BC: What do the dots in these plots show? How is data aggregated? The caption says there are 29k variants. The plots only show ~20 dots. This should be explained in the caption.**

Figures 2BC present the correlation between TraP / GERP++ scores and MAF for 29,985 synonymous variants that create strong cryptic splice sites. In the caption for figure 2BC we now describe that the ~30K synonymous variants were binned into 20 groups (dots representing a group's average MAF) based on 5% incremental score intervals.

- 10) Figure 2DEF: Why are all the trap scores changed from panel to panel? The choices should be explained further in the legend. Coloring of bars also is confusing, as colors don't strictly correspond between panels.**

The caption for figure 2DEF has been updated to explain the choice of TraP thresholds for each analysis, as is described in the corresponding Results section text (Page 9; lines 9 – 19 and Page 10; lines 6 - 12). Regarding the color of the bars, yellow-orange-red colors are consistently adopted to indicate: no TraP score threshold, minimum pathogenic TraP score and high TraP score threshold. The colors are now explained in the updated caption.

11)Your responses to answer my questions on evaluating variants should be discussed in the text/Supplemental text.

We have added the response discussing variant evaluation to the Supplementary Information (SI; Page 6; lines 24 – 33).

12)Bare Figures in the Supplement are not useful for the reader. SI: Supplemental Figures should have self-contained captions (I could not find any) and should be discussed somewhere in the text (either main or SI).

We have added detailed captions to all 25 supplementary figures (revised Supplementary Information). Also, we apologize that while we previously added the text referring to the supplementary figures to the manuscript, we had failed to point to its location in the manuscript in our previous response. We have now also expanded that text in the revised manuscript (Page 24; lines 6 – 23).

13)What is being tested in the Mann Whitney U tests? The hypothesis should be formulated.

This is related to our response for comment #2, above.

14)F1-F20 + Figures S6-S25: The scores should be better explained. What do the X-scales in each figure correspond to?

We have expanded and clarified the text describing the TraP model features (Supplementary Information; Pages 1 – 6). We have also updated the labels for all X-axes in the detailed captions of each supplementary figure (figures S1 – S25).

15)Figure S26: in the response you mention that this is a spearman correlation matrix. This should be mentioned in the caption. Also, over which set is this correlation computed?

The spearman correlation matrix is computed over the ExAC 1.46M synonymous variants dataset. This information is now added to the revised manuscript (Page 24; lines 20 – 23). We have also updated the caption of current figure S25 (previously S26).

16)Figure S6: What is the difference between 1 and 2?

The current figure S5 (previously S6) present the distributions of the Variant Status Factor feature (feature F1). The Variant Status Factor has two possible values: 1 – the variant position resides only within exons or only within introns in all the genes' transcripts, and 2 – the variant position resides in exons in some transcripts and in introns in other transcripts. This information is now added to both the supplementary Information text and the caption of figure S5.

17)Figure S8, S9: X-axis should probably be True/False, instead of 0/1.

Current figures S7 and S8 (previously S8 and S9) have been updated accordingly.

REVIEWERS' COMMENTS:

Reviewer #2 (Remarks to the Author):

I would like to thank the authors for sufficiently addressing most of the reviewers comments, leading to a significantly more concise manuscript that should make a welcome addition to Nature Communications. From a scientific reproducibility standpoint, it is unfortunate that the authors opt to not release the source code for their method to the broader research community.

Assuming that the authors address my minor remaining comments, I do not need to re-review the paper before publication.

Spearman is typically capitalized, as it is a name (e.g. p24, line 730).

SI: Alternative TraP Model:

"The decreased accuracy was due to a decrease in model specificity."

Evidence for this statement should be provided.

Reviewer #2:

1) Spearman is typically capitalized, as it is a name (e.g. p24, line 730).

We have revised the text accordingly.

2) SI: Alternative TraP Model:

"The decreased accuracy was due to a decrease in model specificity."

Evidence for this statement should be provided.

The alternative model with the un-curated variants had a lower specificity than the model with only the curated 75 pathogenic variants, since it miss-classified 8/402 benign training variants as pathogenic compared to only 4/402 with the original model. We have revised the relevant text accordingly.